# Large domains of heterochromatin direct the formation of short mitotic chromosome loops

Maximilian H Fitz-James[1], Pin Tong[1], Alison L Pidoux[1], Hakan Ozadam[2], Liyan Yang[2], Sharon A White[1], Job Dekker[2,3], Robin C Allshire[1]*

[1]Wellcome Centre for Cell Biology and Institute of Cell Biology, School of Biological Sciences, University of Edinburgh, Edinburgh, United Kingdom; [2]Program in Systems Biology, Department of Biochemistry and Molecular Pharmacology, University of Massachusetts Medical School, Worcester, United States; [3]Howard Hughes Medical Institute, Chevy Chase, United States

**Abstract** During mitosis chromosomes reorganise into highly compact, rod-shaped forms, thought to consist of consecutive chromatin loops around a central protein scaffold. Condensin complexes are involved in chromatin compaction, but the contribution of other chromatin proteins, DNA sequence and histone modifications is less understood. A large region of fission yeast DNA inserted into a mouse chromosome was previously observed to adopt a mitotic organisation distinct from that of surrounding mouse DNA. Here, we show that a similar distinct structure is common to a large subset of insertion events in both mouse and human cells and is coincident with the presence of high levels of heterochromatic H3 lysine nine trimethylation (H3K9me3). Hi-C and microscopy indicate that the heterochromatinised fission yeast DNA is organised into smaller chromatin loops than flanking euchromatic mouse chromatin. We conclude that heterochromatin alters chromatin loop size, thus contributing to the distinct appearance of heterochromatin on mitotic chromosomes.

**\*For correspondence:**
robin.allshire@ed.ac.uk

## Introduction

During mitosis, chromatin is dramatically reorganised and compacted to form individual rod-shaped mitotic chromosomes that allow for proper segregation of the genetic material (*Belmont, 2006*). Key to mitotic chromosome compaction are the action of the condensin complexes, condensin I and condensin II, and topoisomerase IIα, whose depletion or deletion can result in defective chromosome compaction and genome stability in a wide range of organisms (*Hirano, 2012*; *Maeshima and Laemmli, 2003*; *Strunnikov et al., 1995*, *Saka et al., 1994*, *Wignall et al., 2003*; *Ono et al., 2003*; *Hirota et al., 2004*; *Martin et al., 2016*; *Woodward et al., 2016*; *DiNardo et al., 1984*; *Uemura et al., 1987*; *Holm et al., 1985*). A leading model for mitotic chromosome compaction proposes that chromatin is organised into loops that radiate out from a core scaffold composed of these key organising complexes (*Maeshima and Eltsov, 2008*). This radial loop model is supported by early cytological and biochemical observations as well as by more recent Hi-C approaches, which in mitosis detect uniform interaction patterns along chromosomes that are consistent with sequence-independent chromatin loops of between 80 and 120 kb (*Paulson and Laemmli, 1977*; *Marsden and Laemmli, 1979*; *Maeshima and Laemmli, 2003*, *Woodcock and Ghosh, 2010*, *Naumova et al., 2013*; *Gibcus et al., 2018*).

The 'loop extrusion model' provides the likely mechanism underlying the formation of mitotic chromosome loops. In this model, condensin complexes extrude loops of chromatin through the centre of their ring-like structures (*Cheng et al., 2015*, *Goloborodko et al., 2016*; *Ganji et al.,*

2018; Gibcus et al., 2018). Both complexes, condensin I and condensin II, are important for mitotic chromosome structure. However, they fulfil different roles. Condensin I drives the axial compaction of the chromosome, and an excess of it leads to longer thinner chromosomes. Conversely, Condensin II is responsible for the shortening and widening of the chromosome and leads to greater axial rigidity (Shintomi and Hirano, 2011; Green et al., 2012; Yamashita et al., 2011; Lai et al., 2011). Nevertheless, many aspects of mitotic chromosome formation, including mechanisms by which it might be modulated to generate distinct cytological structures, remain speculative. Within mitotic chromosomes centromeres, and to a lesser extent telomeres, appear as visually distinct regions of constriction (Belmont, 2006). Studies in both yeast and vertebrates have found that condensin is enriched over these regions (Wang et al., 2005; D'Ambrosio et al., 2008, Kim et al., 2013), potentially explaining these structures as a local alteration in the loop structure. However, the mechanism by which condensin might be recruited or regulated locally in this way is unknown.

One possible explanation is the presence of large domains of constitutive heterochromatin, characterised by the presence of high levels of trimethylation on lysine 9 of H3 (H3K9me3). This type of chromatin is most strongly associated with large domains of repetitive DNA, most notably in pericentromeric regions where it has a vital role in ensuring centromere identity and accurate chromosome segregation (Bernard et al., 2001, Kellum and Alberts, 1995, Shimura et al., 2011, Molina et al., 2016). Many of the roles of H3K9me3 are mediated through its recruitment of HP1 proteins (Kim and Kim, 2012; Lomberk et al., 2006) which act partly by recruiting chromatin modifiers, thus reinforcing and spreading the heterochromatin domain, as well as leading to the compaction of the chromatin fibre (Allshire and Madhani, 2018; Schotta et al., 2004; Fuks et al., 2003; Zhang et al., 2002; Canzio et al., 2011; Martens and Winston, 2003; Lomberk et al., 2006). The majority of HP1 proteins dissociate from chromatin early in mitosis (Fischle et al., 2005; Hirota et al., 2005), however a pool of HP1α remains at centromeres throughout mitosis and is thought to be important for chromosome stability and kinetochore function (Tanno et al., 2015; Abe et al., 2016). Whether the presence of H3K9me3 or HP1α at centromeres is related either directly or indirectly to its structural appearance as the primary constriction in mitosis is not known.

The F1.1 cell line was originally made by fusing fission yeast Schizosaccharomyces pombe spheroplasts, carrying an integrated mammalian selectable marker, with the mouse mammary tumour cell line C127 (Allshire et al., 1987 and McManus et al., 1994). F1.1 carries an insertion of several Mb of S. pombe DNA at a single location on one mouse chromosome. Cytological analysis revealed that the region of the mouse chromosome containing S. pombe DNA adopted a distinct structure in mitosis, manifesting as a region of low DNA staining and apparently narrower diameter (McManus et al., 1994). However, the nature and origin of the structural difference between the inserted S. pombe DNA and the surrounding endogenous mouse DNA was not fully explained. The F1.1 cell line thus represents a useful system for exploring features that locally alter mitotic chromosome structure.

Here, we further investigate the unusual chromatin formed over S. pombe DNA residing within a mouse chromosome in the F1.1 cell line and in several newly-generated cell lines. Through insertion of large regions of S. pombe DNA into mouse NIH3T3 and human HeLa cells by both cell fusion and DNA transfection we conclude that the distinctive chromosome structure previously observed in F1.1 is not unique to a single cell line or species. We show that in various cell lines the inserted S. pombe DNA is packaged into H3K9me3-heterochromatin and that the presence of a sizable block of heterochromatin at the inserted S. pombe DNA correlates with the unusual structure exhibited on metaphase chromosomes. Finally, imaging and Hi-C analyses indicate that the distinct structure is due to altered chromatin organisation and that condensin is enriched over this region. We propose a model whereby elevated condensin association with heterochromatin organises the underlying chromatin into arrays of loops that are smaller than those of surrounding non-heterochromatin regions, thus explaining the observed localised alteration of mitotic chromosome structure.

## Results

### *S. pombe* DNA incorporated into a mouse chromosome adopts a distinct structure with less DNA per unit length

The previously-described F1.1 cell line contains a large stable insertion of *S. pombe* DNA into a single chromosome in mouse C127 (ATCC CRL-1616) cells (*Allshire et al., 1987*). The region containing the integrated fission yeast DNA had a highly distinctive appearance in metaphase spreads (*McManus et al., 1994*). However, the nature of this unusual *S. pombe* DNA-associated chromosome structure and the mechanisms by which it was formed and differentiated from neighbouring mouse chromatin have not been investigated. Recent advances in both our understanding of mitotic chromosome structure and the methodologies used to analyse it now enhance our ability to examine such unusual structures and thereby provide new insight into the processes that govern mitotic chromosome structure.

To confirm and extend earlier analysis, fluorescence in situ hybridisation (FISH) was performed on F1.1 metaphase spreads using a total *S. pombe* genomic DNA probe to decorate the insert. As was previously reported, the region of the mouse chromosome containing *S. pombe* DNA appeared distinct and exhibited reduced DNA staining intensity and a narrower chromosome arm width (*Figure 1A*). Quantification of the DNA and FISH signal intensities along the chromosome length showed a clear decrease in average DNA intensity coinciding with the peak of *S. pombe* DNA FISH signal (*Figure 1B*). This confirmed that the previously reported distinct *Sp*-DNA-associated structure exhibits decreased DNA per unit length of the chromosome compared to the flanking mouse chromatin. The mouse and *S. pombe* genomes have differing GC contents, with 42% for mouse and 36% for *S. pombe* (*Wood et al., 2002*, *Waterston et al., 2002*). To avoid any bias that might result from difference in base composition propidium iodide (PI) staining was used for all DNA intensity measurements rather DAPI, which preferentially binds to AT-rich DNA.

To allow more detailed analysis of the *S. pombe* DNA insert, whole genome sequencing was performed on F1.1 genomic DNA. Initial sequencing by standard next generation sequencing methods revealed that the *S. pombe* DNA was highly rearranged with respect to the *S. pombe* reference genome (assembly ASM294v2). Approximately 64% of the *S. pombe* genome was found to be present within the F1.1 genome, however the high level of rearrangement meant that no assembly could be made to determine the actual sequence of the inserted DNA. Ultra-long sequence reads obtained using nanopore technology allowed assembly of several large contigs of *S. pombe* sequence, including eight contigs greater than 500 kb and one of approximately 1.5 Mb consisting of 1.1 Mb of *S. pombe* DNA extending into 400 kb of adjacent mouse DNA. These results confirmed that the F1.1 insert is composed of rearranged *S. pombe* DNA in which 10 to 100 kb stretches (median length 20 kb) are contiguous with the *S. pombe* reference genome. In addition to providing sequence information crucial to further analysis, our findings suggest that chromatin introduced from yeast by spheroplast fusion is not protected from high levels of rearrangement upon entry into host mammalian cells.

### Chromatin density is lower at the region of mouse chromosome 10 containing *S. pombe* DNA

Both sequencing methods used detected hybrid reads containing both mouse and *S. pombe* DNA sequence and identified the same insertion site on mouse chromosome 10, approximately two-thirds down the length of the chromosome arm (GRCm38 Mm10:83,349,000), consistent with both the size of the insert-bearing mouse chromosome and the relative position of the *Sp*-DNA insert on that mouse chromosome arm (*Figure 1A*).

Two-colour FISH using a mouse chromosome 10 (Mm10) paint along with the total *S. pombe* DNA probe confirmed that in F1.1 cells the *Sp-DNA* insert is carried on one of the copies of chromosome 10 (*Figure 1—figure supplement 1A*). When only the Mm10 paint was employed the *Sp-DNA* insert was clearly visible as a gap in the FISH signal (*Figure 1—figure supplement 1B,C*). Intensity measurements detected the same drop in DNA signal intensity across the *Sp-DNA* insert, in contrast to the other copies of chromosome 10 present in the same cells that exhibit uniform DNA staining along their length (*Figure 1—figure supplement 1C–F*). These analyses exclude the possibility that

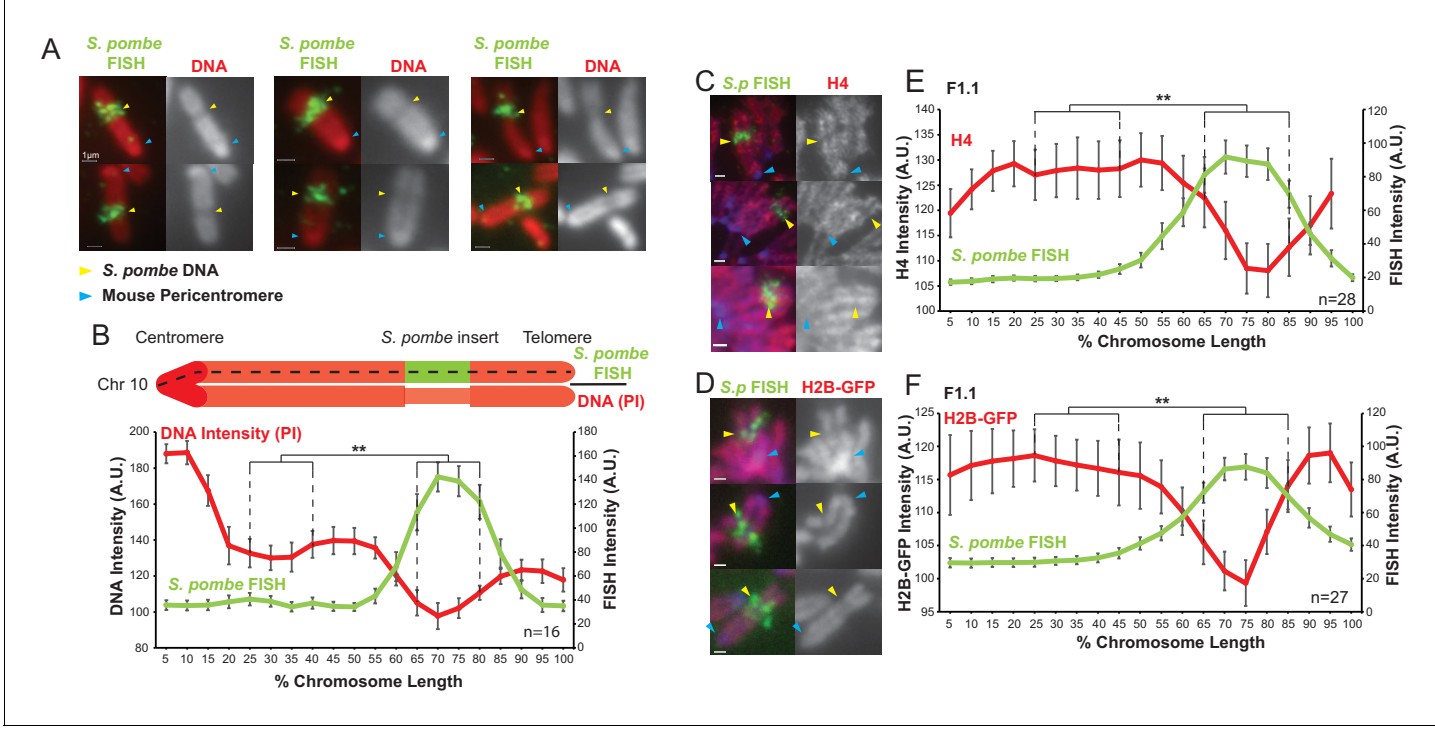

**Figure 1.** *S. pombe* DNA inserted into a mouse chromosome adopts a distinct structure on mitotic chromosomes. (**A**) Metaphase spreads of mouse F1.1 chromosomes showing the distinct structure of the *S. pombe* DNA insert. Propidium iodide stained DNA (PI, red), *S. pombe* DNA FISH using probes from total *S. pombe* DNA (green – yellow arrows), centromeres (regions of brighter DNA staining - blue arrows). Scale bars: 1 μm. (**B**) Schematic representation and average chromosome profile of the F1.1 insert-bearing chromosome across several images (n = 16, *Figure 1—source data 1*). Signal intensities of PI DNA stain (red) and FISH signal (green) were measured along the length of the chromosomes and binned according to their position, from the centromere (0–5%) to the telomere (95–100%). Error bars represent ± standard error from the mean (SEM). Average DNA stain intensity was compared between the regions of 25–40% (endogenous mouse DNA) and 65–80% (*S. pombe* DNA corresponding to the highest FISH signal) by the KS test (**p<0.001). (**C and D**) Metaphase spreads of F1.1 cells either stained by immunofluorescence for histone H4 (**C**) or expressing tagged histone H2B-GFP (**D**) (red), with *S. pombe* DNA FISH (green) and DAPI-stained DNA (blue). *S. pombe* DNA (yellow arrows) and centromere (blue arrows) locations are indicated. Scale bars: 1 μm. (**E and F**) Average signal intensity profile of the F1.1 insert-bearing chromosome showing FISH and either anti-H4 (**E**, *Figure 1—source data 2*) or H2B-GFP (**F**, *Figure 1—source data 3*) across several images (n = 28, 27). Error bars represent ± SEM. Average histone levels were compared between the regions of endogenous mouse DNA and *S. pombe* DNA highlighted by FISH by the KS test (**p<0.001).

The online version of this article includes the following source data and figure supplement(s) for figure 1:

**Source data 1.** FISH and PI intensity measurements for *Figure 1B*.
**Source data 2.** FISH and anti-H4 intensity measurements for *Figure 1E*.
**Source data 3.** FISH and H2B-GFP intensity measurements for *Figure 1F*.
**Figure supplement 1.** The distinct chromatin structure of the *S. pombe* DNA insert on chromosome 10 is not an artefact of the FISH procedure.
**Figure supplement 1—source data 1.** FISH and PI intensity measurements for *Figure 1—figure supplement 1E*.
**Figure supplement 1—source data 2.** FISH and PI intensity measurements for *Figure 1—figure supplement 1F*.

the distinct *Sp-DNA* insert appearance is an artefact of hybridisation with the *S. pombe* DNA FISH probe.

Despite the shorter nucleosome repeat length of chromatin in fission yeast cells (160 bp) compared to that in mammalian cells (185 bp), previous analysis had shown that the *S. pombe* DNA in F1.1 adopts the nucleosome repeat length typical of mouse cells (*McManus et al., 1994*). To assess the density of nucleosomal chromatin across the *Sp-DNA* insert independently of DNA staining immunolocalisation of histone H4 was performed on F1.1 chromosomes. Signal intensity measurements showed a clear drop in H4 signal across the *Sp-DNA* insert. A similar decrease in signal was observed across the *Sp-DNA* insert for ectopically expressed GFP-tagged H2B (*Figure 1C–F*). These analyses indicate that the *Sp-DNA* insert on mouse chromosome 10 of F1.1 cells adopts a distinctive

structure in mitosis that is characterised by a decrease in the amount of chromatin per unit length compared to surrounding endogenous mouse chromatin.

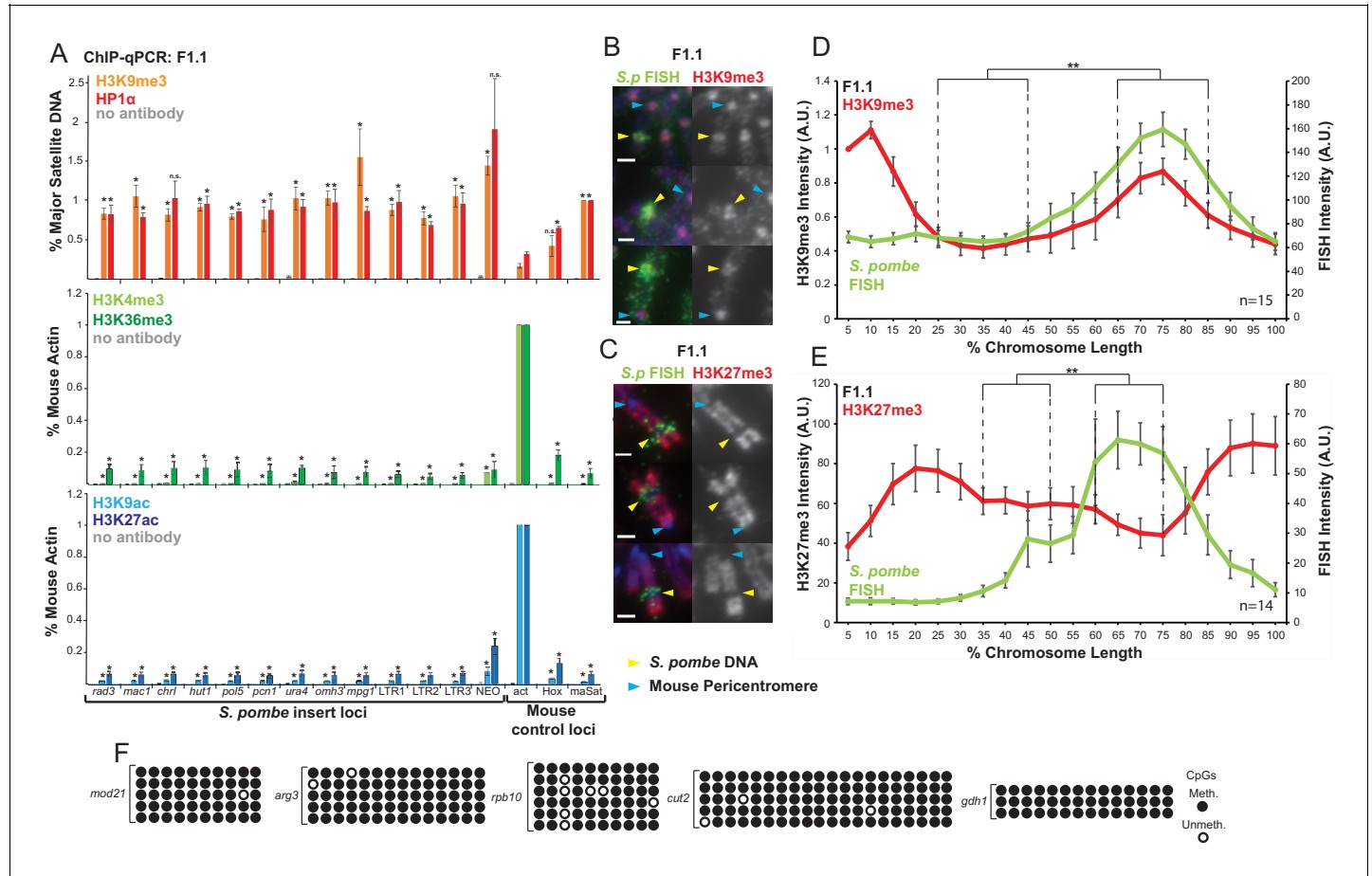

**Figure 2.** *S. pombe* DNA in chromosome 10 of F1.1 is coated in H3K9me3-heterochromatin. (**A**) ChIP-qPCR of F1.1 interphase cells for the repressive marks H3K9me3 and HP1α (top), activating histone methylation marks H3K4me3 and H3K36me3 (middle) and activating acetylation marks H3K9ac and H3K27ac (bottom) at 13 loci within the *S. pombe* insert and three mouse control loci. act - highly transcribed control gene *Actb*; Hox – *Hoxc8* region of facultative heterochromatin; maSat - constitutively heterochromatic centromeric major satellite region. Data in *Figure 2—source data 1*. Error bars represent ± SEM of three independent repeats. Enrichments were normalised to positive control levels and compared to act by the t-test (*p<0.05, n.s. = not significant). (**B,C**) Immunofluorescence for H3K9me3 (**B**) or H3K27me3 (**C**) (red) on F1.1 metaphase spreads showing high H3K9me3 and low H3K27me3 over the *S. pombe* DNA insert as visualised by FISH (green), with DAPI-stained DNA (blue). *S. pombe* DNA (yellow arrows) and centromere (blue arrows) locations are indicated. Scale bars: 1 μm. (**D,E**) Average FISH and H3K9me3 (**D** *Figure 2—source data 2*) or H3K27me3 (**E** *Figure 2—source data 3*) signal intensity profiles of the insert-bearing chromosome of F1.1 across several images (n = 15, 14). Error bars represent ± SEM. H3K9me3 intensity levels were normalised to the 0–5% region of the chromosome, corresponding to the acrocentric mouse centromere. Average immunofluorescence intensity was compared between the regions of endogenous mouse DNA and *S. pombe* DNA highlighted by FISH by the KS test (**p<0.001). (**F**) CpG methylation levels at five *S. pombe* loci within the F1.1 insert as determined by bisulfite sequencing. Loci were sequenced in 3 to 6 replicates, with each replicate shown. Circles represent methylated (black) and unmethylated (white) CpGs, respectively.

The online version of this article includes the following source data and figure supplement(s) for figure 2:

**Source data 1.** ChIP results for H3K9me3, HP1α, H3K4me3, H3K36me3, H3K9ac and H3K27ac on F1.1 cells fore *Figure 2A*.

**Source data 2.** FISH and anti-H3K9me3 intensity measurements for *Figure 2D*.

**Source data 3.** FISH and anti-H3K9me3 intensity measurements for *Figure 2E*.

**Figure supplement 1.** Heterochromatin is confined to the region of foreign *S. pombe* DNA in F1.1 cells.

**Figure supplement 1—source data 1.** ChIP results for H3K9me3 on F1.1 and C127 cells for *Figure 2—figure supplement 1*.

## *S. pombe* DNA integrated in mouse chromosome 10 is assembled in H3K9me3-heterochromatin

The contribution of histone modifications to mitotic chromosome structure is a factor that has long been proposed but is not well defined. A number of histone modifications are enriched in mitotic chromatin and chromatin isolated from mitotic cells has been shown to undergo a certain level of compaction even in the absence of condensins and topoisomerases, suggesting that certain modifications may contribute to mitotic chromatin compaction (*Zhiteneva et al., 2017*; *Goto et al., 2002*; *McManus et al., 2006*, *Park et al., 2011*). However, the mechanism by which this compaction may be achieved is unknown.

To investigate the histone modifications present at the *Sp-DNA* insert in F1.1 cells we employed ChIP-qPCR with antibodies detecting H3K4me3, H3K9ac, H3K9me3, H3K27ac and H3K36me3. Expected enrichments of histone modifications associated with active chromatin, such as H3K4me3, H3K9ac, H3K27ac and H3K36me3 were detected on the mouse *Actb* gene and were low on the repressed *Hoxc8* gene and mouse major satellite repeats. In contrast, levels of the heterochromatic H3K9me3 were high on mouse major satellite repeats but low on *Actb*. Active modifications were also enriched on the selected G418 resistance gene SV40-Neo, present within the *Sp-DNA* insert, consistent with its expression. Strikingly, all twelve *S. pombe* DNA sites examined within the *Sp-DNA* insert exhibited very high levels of H3K9me3 and low levels of H3K4me3, H3K9ac, H3K27ac and H3K36me3 relative to the mouse *Actb*. High levels of HP1α, which binds H3K9me3, were also detected at all locations tested within the F1.1 *Sp-DNA* insert (*Figure 2A*). The presence of both H3K9me3 and HP1α across the *Sp-DNA* insert suggests that a large domain of constitutive heterochromatin coats this foreign DNA.

This was supported by immunolocalisation, which showed a high intensity of H3K9me3 coinciding with the *S. pombe* DNA on metaphase spreads. In addition, H3K27me3 was low over the *S. pombe* DNA, demonstrating that the fission yeast DNA was assembled into constitutive H3K9me3-dependent heterochromatin rather than facultative H3K27me3-dependent heterochromatin (*Figure 2B–E*). Mammalian constitutive heterochromatin is also known to be highly methylated on CpG dinucleotides (*Bannister and Kouzarides, 2011*; *Rose and Klose, 2014*, *Bogdanović and Veenstra, 2009*). Consistent with methylation-sensitive restriction enzyme digestion (*McManus et al., 1994*), bisulphite sequencing demonstrated prevalent CpG DNA methylation within the *Sp-DNA* insert (*Figure 2F*). Additional H3K9me3 ChIP analysis demonstrated that this heterochromatin did not encroach on nearby mouse DNA flanking the site of *Sp-DNA* insertion, and is therefore confined to the foreign DNA (*Figure 2—figure supplement 1*). We conclude that constitutive heterochromatin coats the exogenous *Sp-DNA* insert on mouse chromosome 10.

## A distinct heterochromatic structure frequently forms on *S. pombe* DNA integrated into mouse or human chromosomes

The formation of a distinct structure assembled into heterochromatin on *S. pombe* DNA inserted in mouse chromosome 10 might result from unique events associated with the specific integration site in F1.1 cells, the configuration of the inserted *S. pombe* DNA or the mouse C127 cell line. To exclude these possibilities, we fused spheroplasts of the same *S. pombe* strain harbouring the SV40-Neo G418 selectable marker with either mouse NIH3T3 cells or human HeLa cells as previously described (*Allshire et al., 1987*). Subsequent screening of resulting G418-resistant clones by FISH identified one HeLa (HeP-F3) and two NIH3T3 (NP-F1 and NP-F2) clones with large blocks of *S. pombe* DNA integrated at single chromosomal locations. In these three new cell lines, PI staining of mitotic chromosomes again revealed a less intensely stained region coincident with the resulting inserted *S. pombe* DNA (*Figure 3A–F*). The presence of an unusual mitotic chromosome structure on *S. pombe* DNA inserted in four independent fusion clones indicates that the formation of such entities was not a unique event confined to the original F1.1 isolate, the specific cell line used or its species of origin. Moreover, since mitotic chromosomes were examined from the new clones within 25 divisions of their formation these structures are not the consequence of prolonged propagation in cell culture, but must be formed soon after integration of the incoming foreign DNA into a host chromosome. ChIP-qPCR and immunolocalisation demonstrated that, as in F1.1 cells, H3K9me3 was strongly enriched on the *Sp-DNA* inserts in the new NP-F1, NP-F2 and HeP-F3 cell lines (*Figure 3G–I* and *Figure 3—figure supplement 1*). Taken together, these results suggest that, in addition to

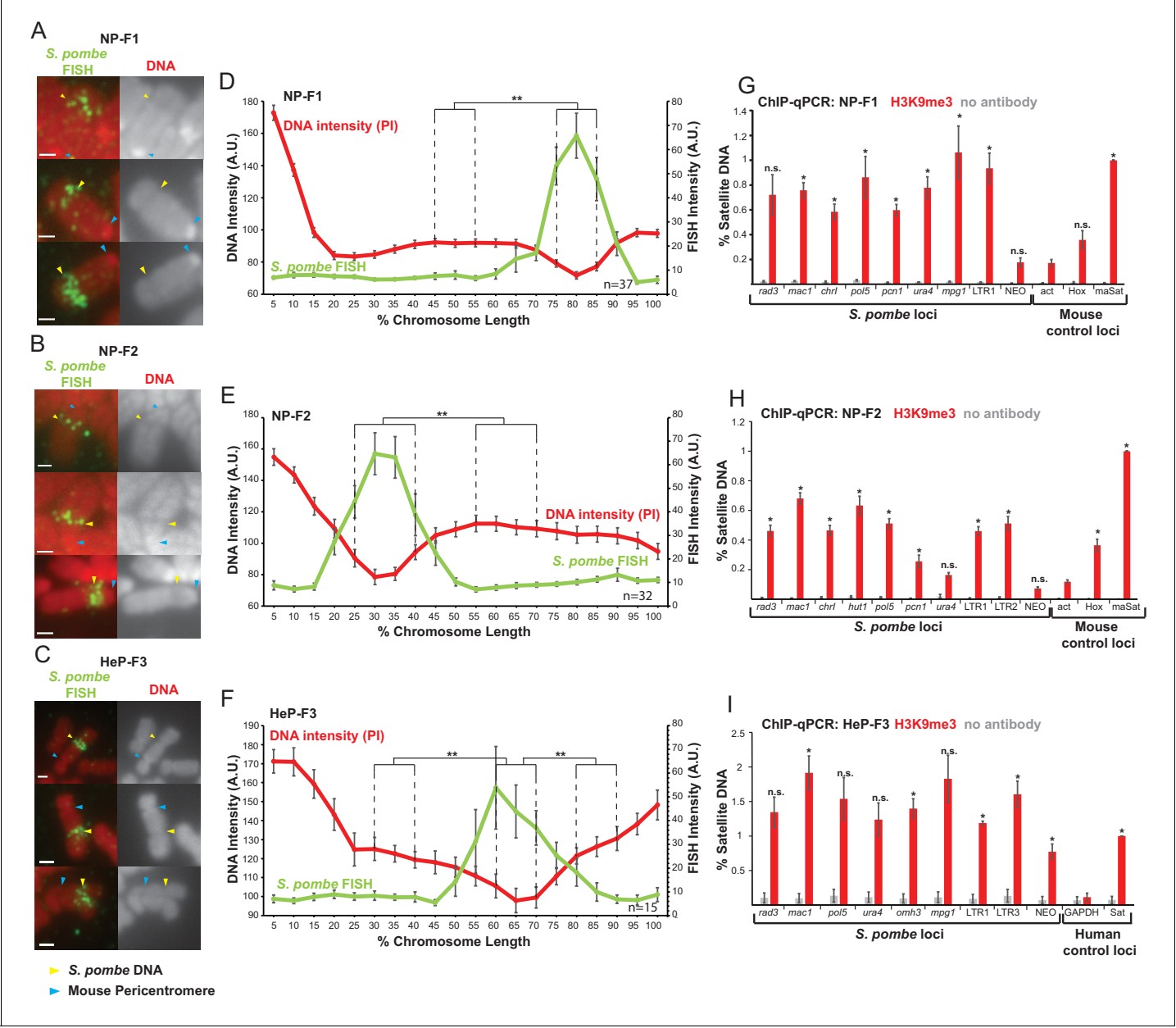

**Figure 3.** *S. pombe* DNA assembles heterochromatin and adopts a distinct mitotic chromatin structure when inserted into mammalian chromosomes. (A–C) Metaphase spreads of NP-F1, NP-F2 and HeP-F3 cells showing the distinct structure over the inserted *S. pombe* DNA. PI-stained DNA (red), *S. pombe* DNA FISH (green – yellow arrows). Blue arrows indicate centromeres. Scale bar: 1 μm. (D–F) Average FISH and PI signal intensity profiles of the insert-bearing chromosomes in each cell line across several images (n = 37, 32 and 15, *Figure 3—source datas 1–3*). Error bars represent ± standard error from the mean (SEM). Average DNA stain intensity was compared between the regions of endogenous mammalian DNA and *S. pombe* DNA highlighted by FISH by the KS test (**p<0.001). (G–I) ChIP-qPCR for H3K9me3 in NP-F1 (G, *Figure 3—source data 4*), NP-F2 (H, *Figure 3—source data 5*) and HeP-F3 (I, *Figure 3—source data 6*) in interphase. qPCR was performed with primers used for analysis of F1.1 (see *Figure 2*) that were present in the retained *S. pombe* DNA. Mouse control regions were: negative control *Actb* gene – <u>act</u>; facultative heterochromatin region – <u>Hox</u>; centromeric heterochromatin major satellite positive control – <u>maSat</u>. Human control regions were: the highly transcribed negative control gene – <u>GAPDH</u>; human alpha satellite DNA positive control – <u>Sat</u>. Error bars represent ± SEM of three independent repeats. Enrichments were normalised to positive control levels and compared to the negative control by the t-test (*p<0.05, n.s. = not significant).

The online version of this article includes the following source data and figure supplement(s) for figure 3:

**Source data 1.** FISH and PI intensity measurements for *Figure 3D*.
**Source data 2.** FISH and PI intensity measurements for *Figure 3E*.
**Source data 3.** FISH and PI intensity measurements for *Figure 3F*.
**Source data 4.** ChIP results for H3K9me3 on NP-F1 cells for *Figure 3G*.

*Figure 3 continued on next page*

*Figure 3 continued*

**Source data 5.** ChIP results for H3K9me3 on NP-F2 cells for *Figure 3H*.
**Source data 6.** ChIP results for H3K9me3 on HeP3 cells for *Figure 3I*.
**Figure supplement 1.** Heterochromatin assembled on *S.pombe* DNA inserted into NIH3T3 chromosomes is visible by microscopy.
**Figure supplement 1—source data 1.** FISH and anti-H3K9me3 intensity measurements for *Figure 3—figure supplement 1C*.
**Figure supplement 1—source data 2.** FISH and anti-H3K9me3 intensity measurements for *Figure 3—figure supplement 1D*.

having a distinct mitotic structure, the *S. pombe* DNA has been assembled into H3K9me3-dependent heterochromatin in all four mammalian-*S. pombe* fusion hybrids.

## Heterochromatin forms on incoming *S. pombe* chromatin that lacks pre-existing H3K9 methylation

The *S. pombe* genome contains blocks of H3K9 methylation at least every 4 Mbp: at the three centromeres, six telomeres and the mating type locus. The accumulation of high levels of H3K9me3 on *S. pombe* DNA inserted into mammalian chromosomes following spheroplast fusion in the F1.1, NP-F1, NP-F2 and HeP-F3 cell lines might result from these pre-existing blocks of H3K9 methylation in *S. pombe* acting as nucleation sites from which heterochromatin spreads over integrated *S. pombe* DNA. Alternatively, this heterochromatin may simply be a default state formed on large blocks of essentially inert foreign DNA. Clr4 is the only H3K9 methyltransferase encoded by the *S. pombe* genome. It is not essential for viability and yeast cells lacking Clr4 are devoid of H3K9me2/3 and heterochromatin (*Nakayama et al., 2001*). We therefore prepared spheroplasts from *S. pombe* cells lacking Clr4 (*clr4Δ*) and carrying the same SV40-Neo selectable marker, fused them with NIH3T3 cells and selected G418-resistant hybrids. The resulting cell line NP-*clr4Δ*-F4 was found to contain a single large *S. pombe* DNA insertion (*Figure 4—figure supplement 1A*) which exhibited reduced PI staining intensity on metaphase chromosomes (*Figure 4A,B*, *Figure 4—figure supplement 1B*). Surprisingly, ChIP and immunolocalisation showed that H3K9me3 heterochromatin had also been formed over the inserted *S. pombe* DNA even in this NP-*clr4Δ*-F4 cell line (*Figure 4C–E*).

We conclude that the establishment of heterochromatin on *S. pombe* DNA inserted into mammalian chromosomes following spheroplast fusion is not dependent on the pre-existing domain of H3K9 methylation on *S. pombe* chromosomes. Thus, unhalted spreading from such regions does not explain the formation of these extensive structures. It seems likely that heterochromatin formation is triggered de novo after delivery of *S. pombe* chromatin into mammalian cells.

## Large *S. pombe* DNA insertions introduced as transfected naked DNA can form distinct heterochromatic structures in metaphase chromosomes

The above analyses exclude the possibility that the unusual mitotic chromosome structure formed on inserted *S. pombe* DNA is dependent on resident H3K9 methylation. However, it is possible that some other feature carried by incoming *S. pombe* chromatin, and retained on chromatin from *clr4Δ* cells, promotes heterochromatin formation. The calcium-phosphate precipitation DNA transfection method is known to allow the assembly of large 'transgenomes' which tend to integrate at single chromosomal locations in mammalian cells (*Perucho et al., 1980*; *Scangos et al., 1981*). Therefore, we used calcium-phosphate precipitation to transfect NIH3T3 cells with large amounts of *S. pombe* genomic DNA, free from all chromosomal proteins and prepared from cells carrying the SV2-Neo selectable marker. Resulting G418[R] transformants were screened by *S. pombe* DNA FISH. Six cell lines (NP-T1 to NP-T6) were isolated that contained large *S. pombe* insertions at a single chromosomal location (*Figure 5A–F*).

Analysis of metaphase chromosomes containing *S. pombe* DNA insertions in these transformants revealed that their appearance varied considerably. In NP-T1 and NP-T2 there was no significant detectable difference in chromosome structure or PI intensity over the region containing *S. pombe* DNA compared to the rest of the chromosome arm (*Figure 5A–B,G–H*). On a proportion of chromosomes from NP-T3 a slightly different structure and noticeably decreased PI intensity was observed to coincide with the *S. pombe* DNA (*Figure 5C*), although this difference was only statistically significant when compared with one of the two control regions (*Figure 5I*). In contrast, NP-T4, NP-T5 and

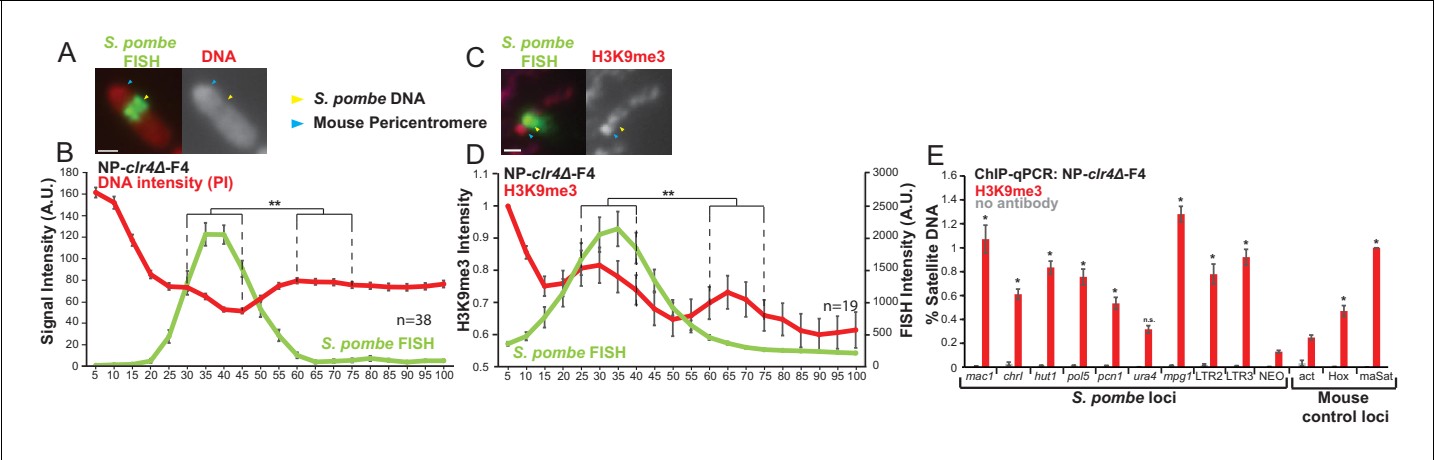

**Figure 4.** *S. pombe* DNA from cells lacking H3K9 methylation forms heterochromatin and adopts a distinct structure when incorporated into mammalian chromosomes. (**A**) Metaphase spread of NP-*clr4Δ*-F4 showing the distinct structure over the *S. pombe* DNA insert, with PI-stained DNA (red) and *S. pombe* DNA FISH (green – yellow arrows). Blue arrows indicate the centromere. Scale bar: 1 μm. (**B**) Average FISH and DNA stain (PI) signal intensity profile of the insert-bearing chromosomes of NP-*clr4Δ*-F4 (n = 38, *Figure 4—source data 1*). Error bars represent ± SEM. Average DNA stain intensity was compared between a region of endogenous mouse DNA and *S. pombe* DNA highlighted by FISH by the KS test (\*\*p<0.001). (**C**) Immunofluorescence for H3K9me3 (red) on a NP-*clr4Δ*-F4 metaphase spread with DAPI-stained DNA (blue) and *S. pombe* DNA FISH (green - yellow arrow). Blue arrows indicate the centromere. Scale bars: 1 μm. (**D**) Average FISH and H3K9me3 signal intensity profile along the NP-*clr4Δ*-F4 insert-bearing chromosome from several images (n = 19, *Figure 4—source data 2*). Error bars represent ± SEM. H3K9me3 intensity levels were normalised to the 0–5% region of the chromosome, corresponding to the acrocentric mouse centromere. Average H3K9me3 intensity was compared between the regions of endogenous mouse DNA and *S. pombe* DNA highlighted by FISH by the KS test (\*p<0.01). (**E**) ChIP-qPCR on NP-*clr4Δ*-F4 interphase cells for H3K9me3 at several loci within the *S. pombe* insert and three mouse control loci. Negative control gene *Actb* - <u>act</u>, facultative heterochromatin - <u>Hox</u>, centromeric heterochromatn major satellite positive control – <u>maSat</u>. Data in *Figure 4—source data 3*. Error bars represent ± SEM of three independent repeats. Enrichments were normalised to <u>maSat</u> levels and compared to <u>act</u> by the t-test (\*p<0.05, n.s. = not significant).

The online version of this article includes the following source data and figure supplement(s) for figure 4:

Source data 1. FISH and PI intensity measurements for *Figure 4B*.
Source data 2. FISH and anti-H3K9me3 intensity measurements for *Figure 4D*.
Source data 3. ChIP results for H3K9me3 on NP-*clr4Δ*-F4 cells for *Figure 4E*.
Figure supplement 1. Distinct structure formed by DNA from *S. pombe clr4Δ* cells when inserted into a mouse NIH3T3 chromosome.

NP-T6 all displayed visibly distinct structures on the majority of mitotic chromosomes carrying *S. pombe* DNA, with NP-T6 exhibiting the most obvious distinct appearance (*Figure 5D–F*). PI intensity was found to decrease significantly over the chromosomal site of *S. pombe* DNA insertion in these three DNA transformants (*Figure 5J–L*).

H3K9me3 levels at the inserted *S. pombe* DNA were similarly found to vary between these six cell lines and, strikingly, this variation correlated with the distinct appearance of the *S. pombe* DNA. NP-T1 and NP-T2 had relatively low H3K9me3 levels compared to the centromeric satellite positive control region (30–35% of the signal associated with the major satellite positive control) (*Figure 5M, N*), NP-T3 exhibited intermediate levels of H3K9me3 levels on its *S. pombe* DNA insert (67% of the major satellite signal) (*Figure 5O*) whereas considerably higher H3K9me3 levels were associated with the insertions in NP-T4, NP-T5 and NP-T6 (80–150% of the major satellite signal) (*Figure 5P–R*). Immunolocalisation confirmed the above variability: NP-T1 and NP-T2 showed no visible H3K9me3 signal over the inserted *S. pombe* DNA. A low but significant increase in H3K9me3 signal was detectable over the inserted DNA in NP-T3 and NP-T4, while obvious domains of H3K9me3 were observed over the *S. pombe* DNA in NP-T5 and NP-T6 (*Figure 5—figure supplement 1*). Thus, *S. pombe* DNA insertion events exhibiting the most evident and distinct structural appearance by PI staining are associated with the highest levels of H3K9 methylation. This association suggests that the assembly of heterochromatin on inserted *S. pombe* DNA is responsible for the distinct appearance.

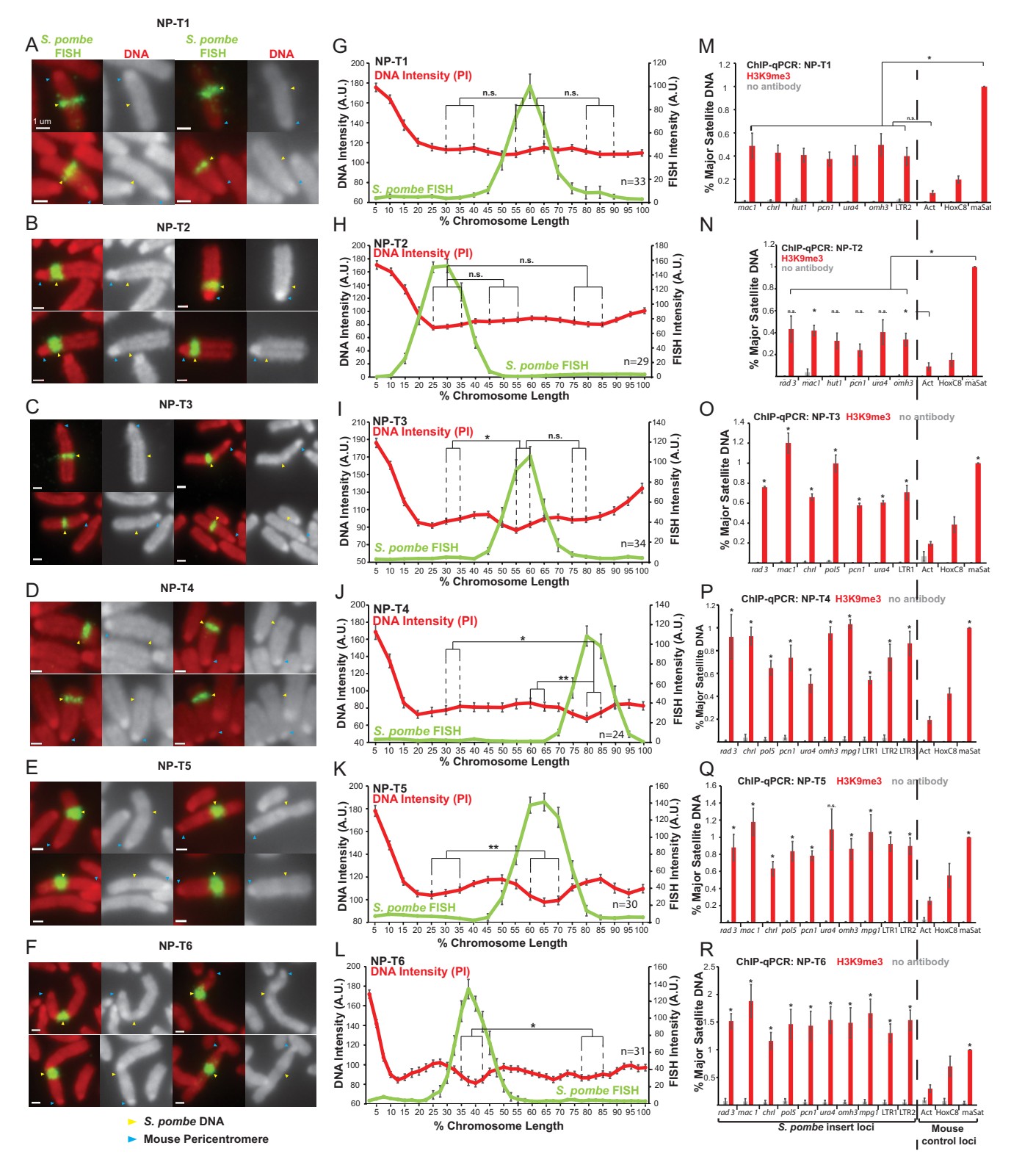

**Figure 5.** Distinct metaphase structure correlates with H3K9me levels over regions of *S. pombe* DNA inserted in mammalian chromosomes. (**A–F**) Metaphase spreads of *S. pombe* DNA insert-bearing chromosomes from *S. pombe* DNA transfected cell lines NP-T1, T2, T3, T4, T5 and T6. PI-stained DNA (red), *S. pombe* DNA FISH (green - yellow arrows), centromeres (blue arrows). Scale bars: 1 μm. (**G–L**) Average FISH and DNA stain (PI) signal intensity profiles along *S. pombe* DNA insert-bearing chromosomes of NP-T1 to NP-T6 (n = 33, 29, 34, 24, 30, 31, *Figure 5—source datas 1–6*). Error

*Figure 5 continued on next page*

*Figure 5 continued*

bars represent ± SEM. Average DNA stain intensity was compared between regions of endogenous mouse DNA and *S. pombe* DNA highlighted by FISH by the KS test (*p<0.01, **p<0.001, n.s. = not significant). (**M–R**) ChIP-qPCR on NP-T1 to NP-T6 interphase cells for H3K9me3 at several loci within the *S. pombe* insert and three mouse control loci. Negative control gene *Actb* - <u>act</u>, facultative heterochromatin - <u>Hox</u>, centromeric heterochromatin major satellite positive control – <u>maSat</u>. Data in *Figure 5—source datas 7–12*. Error bars represent ± SEM of three independent repeats. Enrichments were normalised to maSat levels and compared to Act (and maSat for NP-T1 and NP-T2) by the t-test (*p<0.05, n.s. = not significant).

The online version of this article includes the following source data and figure supplement(s) for figure 5:

**Source data 1.** FISH and PI intensity measurements for *Figure 5G*.
**Source data 2.** FISH and PI intensity measurements for *Figure 5H*.
**Source data 3.** FISH and PI intensity measurements for *Figure 5I*.
**Source data 4.** FISH and PI intensity measurements for *Figure 5J*.
**Source data 5.** FISH and PI intensity measurements for *Figure 5K*.
**Source data 6.** FISH and PI intensity measurements for *Figure 5L*.
**Source data 7.** ChIP results for H3K9me3 on NP-T1 cells for *Figure 5M*.
**Source data 8.** ChIP results for H3K9me3 on NP-T2 cells for *Figure 5N*.
**Source data 9.** ChIP results for H3K9me3 on NP-T3 cells for *Figure 5O*.
**Source data 10.** ChIP results for H3K9me3 on NP-T4 cells for *Figure 5P*.
**Source data 11.** ChIP results for H3K9me3 on NP-T5 cells for *Figure 5Q*.
**Source data 12.** ChIP results for H3K9me3 on NP-T6 cells for *Figure 5R*.
**Figure supplement 1.** Variable levels of H3K9me3 accumulate on *S. pombe* DNA transgenomes in mammalian chromosomes.
**Figure supplement 1—source data 1.** FISH and anti-H3K9me3 intensity measurements for *Figure 5—figure supplement 1D*.
**Figure supplement 1—source data 2.** FISH and anti-H3K9me3 intensity measurements for *Figure 5—figure supplement 1E*.
**Figure supplement 1—source data 3.** FISH and anti-H3K9me3 intensity measurements for *Figure 5—figure supplement 1F*.
**Figure supplement 1—source data 4.** FISH and anti-H3K9me3 intensity measurements for *Figure 5—figure supplement 1J*.
**Figure supplement 1—source data 5.** FISH and anti-H3K9me3 intensity measurements for *Figure 5—figure supplement 1K*.
**Figure supplement 1—source data 6.** FISH and anti-H3K9me3 intensity measurements for *Figure 5—figure supplement 1L*.

## Heterochromatinised *S. pombe* DNA forms smaller chromatin loops in mitosis

To determine how *Sp-DNA* insert-associated H3K9me3-heterochromatin might affect mitotic chromatin structure, we further examined the F1.1 cell line. While heterochromatin is generally thought to compact chromatin fibres in interphase, our analysis indicates the opposite: less chromatin per unit length is detected across the *Sp-DNA* insert on F1.1 mitotic chromosomes (*Figure 1*). A possible explanation is that chromatin fibre organisation, rather than fibre structure itself, is altered in these regions. Current models of mitotic chromosome organisation propose that the chromatin fibre is organised into consecutive loop arrays organised around a central scaffold as originally suggested by observations of extracted chromosomes (*Paulson and Laemmli, 1977*; *Adolph et al., 1977*; *Earnshaw and Laemmli, 1983*). These models are in part based upon Hi-C data which shows that in mitosis mammalian cells lose all Topologically Associating Domain (TAD) structures and chromosome compartments and show a uniform interaction pattern consisting of strong interactions along the diagonal that undergo a sharp drop-off over a longer range. This drop-off was most strikingly visualised by plotting contact probability over genomic distance in a P(s) curve (*Naumova et al., 2013*; *Gibcus et al., 2018*). Polymer simulations demonstrated these findings to be consistent with an organisation of chromatin into arrays of loops. Furthermore, the distance at which the drop-off in interaction frequency occurs, was shown to be indicative of the amount of chromatin per 'layer' of loop arrays. This is explained by the fact that within one layer contact between two loci is very likely as loops are tightly packed and in close proximity. Conversely, contact between two loci separated by a distance greater than one layer is highly restricted and thus unlikely to occur.

To further investigate the organisation of chromatin within the *Sp-DNA* insert we therefore performed Hi-C on mitotic F1.1 cells. Consistent with previous results, F1.1 mitotic cells lose long-range interactions and present a uniform pattern of strong short-range interactions (*Figure 6—figure supplement 1*). We then compared the longest *Sp-DNA* contig assembled from our sequencing of F1.1 with a region of mouse DNA of the same size on chromosome 10 (*Figure 6A,B*). Both regions showed the expected mitotic Hi-C pattern but with different characteristics. While both regions showed strong short-range interactions along the diagonal, the strength of the interactions over the

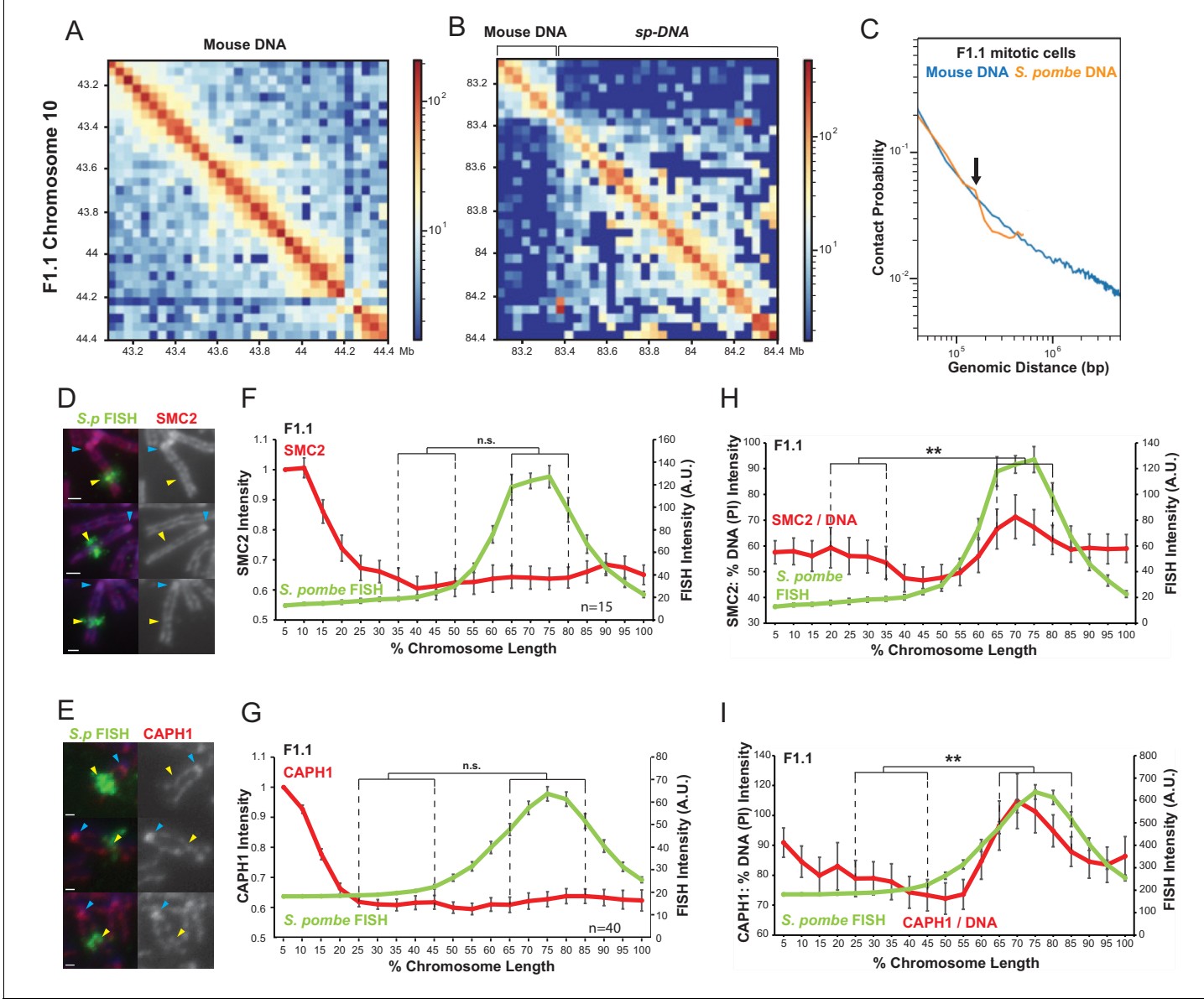

**Figure 6.** Chromatin assembled on *S. pombe* DNA in F1.1 cells is organised into smaller loops than flanking mouse chromatin in mitosis. (**A,B**) Hi-C contact heatmaps for the longest contig containing *S. pombe* DNA and flanking mouse DNA from the F1.1 de novo genome assembly (**B**) and a contiguous region of mouse chromosome 10 of the same size (**A**). (**C**) Chromatin interactions as a function of genomic distance in F1.1 mitotic cells. Contact probability plotted for mouse sequences and *S. pombe* sequences mapped to the F1.1 de novo assembly. Arrow indicates drop in *S. pombe* interactions. (**D,E**) Immunofluorescence for SMC2 (**D**) or CAPH1 (**E**) on F1.1 metaphase spreads (red); *S. pombe* DNA FISH (green - yellow arrows), DAPI-stained DNA (blue), centromeres (blue arrows). Scale bars: 1 μm. (**F,G**) Average FISH and SMC2 (F, *Figure 6—source data 1*) or CAPH1 (G, *Figure 6—source data 2*) signal intensity profiles along the insert-bearing chromosome of F1.1 from several images (n = 15, 40). SMC2 and CAPH1 intensity levels were normalised to the 0–5% region of the chromosome, corresponding to the mouse centromere. (**H,I**) Normalisation of the SMC2 (**H**) and CAPH1 (**I**) profiles to DNA levels as measured by PI intensity. FISH intensities remain unaltered. Error bars represent ± SEM. Values were compared between a region of endogenous mouse DNA and the region of *S. pombe* DNA highlighted by FISH by the KS test (**p<0.001, n.s. = not significant). The online version of this article includes the following source data and figure supplement(s) for figure 6:

**Source data 1.** FISH and anti-SMC2 intensity measurements for *Figure 6F*.
**Source data 2.** FISH and anti-SMC2 intensity measurements for *Figure 6G*.
**Figure supplement 1.** Loss of long-range interphase chromatin contacts in F1.1 mitotic cells.
**Figure supplement 2.** CTCF sites are not enriched over the *Sp-DNA* insert.
**Figure supplement 3.** Condensin levels increase relative to chromatin over the region of *S. pombe* DNA insertion in a mammalian chromosome.

mouse DNA region decreased more gradually with distance within the range observed. Conversely, the *Sp-DNA* region showed a sharper drop-off in interaction frequency. To visualise this more clearly, we made *P*(*s*) curves for these two datasets, plotting the probability of contact as a function of genomic distance for all mouse and *S. pombe* regions on mitotic chromosomes (*Figure 6C*). For *S. pombe* regions we used our de novo assembled contigs from F1.1 sequencing as a reference. However, as few of these contigs are greater than several hundred kb in length, the plot became erratic at large genomic distances due to the paucity of longer contigs. We thus truncated the *S. pombe* interaction plot at 500 kb, the N50 of our dataset. Nevertheless, within the observable range the plot suggests a more rapid decline in interaction frequencies within the *S. pombe* DNA component. Both mouse and *S. pombe* regions show extremely similar interaction patterns at short range, with a gradual decrease in interaction as distance increases, up to approximately 150 kb. However, at a distance of 150 kb interactions within the *S. pombe* region show an abrupt drop while interactions within mouse chromosomal regions decline much more gradually. As explained above, this sharp decrease in contact probability is a clear indication that the *Sp-DNA* insert has less chromatin per layer on mitotic chromosomes than mouse chromosomal regions in the F1.1 cell line. This is both consistent with our microscopy results (*Figure 1*) and suggests that chromatin organisation is indeed altered within the *Sp-DNA* region.

To explore how chromatin might be altered over the *Sp-DNA* insert, and what form this alteration might take, we investigated the possible involvement of proteins known to influence chromatin architecture. First, because CTCF plays a key role in the formation of TADs in interphase in part through the regulation of the loop-extruding activity of cohesin (*Dixon et al., 2012*; *Nora et al., 2017*; *Sanborn et al., 2015*), we performed a search for CTCF consensus binding sites across the *Sp-DNA* insert and neighbouring mouse DNA (*Figure 6—figure supplement 2*). Our analysis demonstrates that the frequency of predicted CTCF binding sites is substantially less within the inserted *S. pombe* DNA, probably because CTCF-binding sites have a strong GC component. This lower frequency of predicted CTCF sites correlates with the decrease in the overall GC content of *S. pombe* DNA, thus the observed structure with smaller loops across inserted S. pombe DNA cannot be explained by a higher predicted frequency of CTCF sites within that DNA. Moreover, mounting evidence indicates that CTCF is not involved in the shaping of chromatin during mitosis (*Oomen et al., 2019*). We therefore conclude that CTCF binding density does not explain the altered chromatin organisation over the *Sp-DNA* insert on mitotic chromosomes.

The condensin complexes are major architects of mitotic chromosome structure and key components of the protein scaffold around which loop arrays are organised (*Hirano, 2012*; *Gibcus et al., 2018*). This is achieved in part through their role as the primary drivers of loop formation in mitosis by loop extrusion (*Goloborodko et al., 2016*; *Ganji et al., 2018*; *Gibcus et al., 2018*). Changes in the activity of condensin or in its association with chromatin can lead to dramatic changes in mitotic chromosome structure (*Wignall et al., 2003*; *Ono et al., 2003*; *Woodward et al., 2016*), although such changes have only been reported globally for all chromosomes, not locally within a specific chromosomal region.

We therefore investigated the association of SMC2 - a core component of both condensin I and condensin II - with F1.1 mitotic chromosomes (*Hirano, 2012*). Unlike histone H2B, H4 and H3K9me3, immunolocalisation showed that SMC2 was uniformly distributed along the length of the entire chromosome arm containing the *Sp-DNA* insert (*Figure 6D,F*). Condensin I and condensin II have different roles in shaping mitotic chromosomes and changes in their relative ratio may alter mitotic chromosome morphology (*Antonin and Neumann, 2016*; *Kschonsak and Haering, 2015*; *Shintomi and Hirano, 2011*; *Lai et al., 2011*; *Yamashita et al., 2011*; *Green et al., 2012*). However, CAPH1, a condensin I-specific component (*Hirano, 2012*), showed a similar even distribution along the entire length of the F1.1 mitotic chromosomes containing the *Sp-DNA* insert (*Figure 6E,G*). Measurement of a condensin II-specific subunit would be desirable to fully demonstrate any change, or lack thereof, in the condesin I/II ratio. However, attempts to image a condensin II-specific subunit on the F1.1 mitotic chromosome were unsuccessful. Nevertheless, the overall condensin levels, as measured by SMC2, and condensin I levels, as measured by CAPH1, are sufficient to infer a condensin I:condensin II ratio. Indeed, no change is observed in either the overall levels of condensin I and II combined, or the level of condensin I alone over the *Sp-DNA* insert. We therefore deduce that, within the limits of quantification, condensin II levels also remain relatively unchanged across the *Sp-DNA* insert compared to the host chromosome 10 in which it is embedded.

These observations therefore suggest that, while DNA and histone levels are lower over the F1.1 *Sp-DNA* insert than in surrounding mouse chromatin, neither the overall levels of condensin nor the ratio of condensin I to condensin II complexes differ dramatically. This indicates that overall there is a localised increase in condensin levels relative to the amount of chromatin across the *Sp-DNA* insert. This increase in the condensin:chromatin ratio is evident when SMC2 or CAPH1 levels are normalised to DNA (*Figure 6H,I*) or H2B (*Figure 6—figure supplement 2*). Based on these observations we propose that the *Sp-DNA* insert is organised into smaller loops of chromatin than the surrounding mouse DNA. As the main player in the formation of mitotic chromatin loops, an increased association of condensin with this region of the chromosome would be expected to result in arrays of smaller loops due to the localised increase in loop-extruding activity. A smaller loop organisation would also result in a decrease in chromatin per unit length of the chromosome (assuming a constant number of condensins per unit length of the chromosome, consistent with the even distribution of condensins along the chromosome carrying the *Sp-DNA* insert). A decrease in the frequency of long-range chromatin contacts for loci separated by more than 150 kb as observed for the *Sp-DNA* indicates a reduced amount of DNA per layer of loops, which would be expected when loops are smaller. Finally, such an organisation would explain the distinct narrow appearance of the *Sp-DNA* insert within mouse chromosome 10. We therefore propose that the heterochromatinised *S. pombe* region has an altered mitotic chromatin loop organisation with a decreased loop size, resulting in a localised narrowing of the chromosome width and reduction in the amount of chromatin per unit length of chromosome.

## Discussion

### Heterochromatin is established over foreign *S. pombe* DNA in mammalian cells

Since its initial description, the presence of a large region of distinct mitotic chromosome structure in the F1.1 cell line has remained an interesting but largely unexplained observation (*Allshire et al., 1987*; *McManus et al., 1994*). By expanding this observation to other cell lines and even another mammalian species, we have shown that this variation in structure is not unique to the F1.1 cell line. This is in agreement with other studies that have reported similar narrow appearances on mitotic chromosomes of DNA inserted into mammalian cells from another yeast species, *S. cerevisiae* (*Featherstone and Huxley, 1993*; *Nonet and Wahl, 1993*).

Our analyses suggest that the observed distinct appearance is related to the assembly of high levels of H3K9me3-heterochromatin over the incoming foreign DNA. This is consistent with the previous observation that the *S. pombe* chromatin in F1.1 is replicated late in S phase (*McManus et al., 1994*). The de novo establishment of a heterochromatin domain over DNA derived from *S. pombe clr4Δ* cells that lack H3K9 methylation led us to reject the hypothesis that heterochromatin was spreading from pre-existing heterochromatin carried on incoming *S. pombe* chromatin (*Figure 4*). Furthermore, heterochromatin was also established to varying degrees over *S. pombe* DNA introduced by transfection as naked DNA into mammalian cells (*Figure 5*), precluding any possible influence of *S. pombe* chromatin on the inserted domain.

However, it remains possible that the spreading of heterochromatin over the inserted *S. pombe* DNA in a mammalian cell could happen quite readily. Over endogenous chromatin, heterochromatin spreading is usually checked by barriers, such as regions of active transcription, active histone modifications and high nucleosome turnover (*Allshire and Madhani, 2018*). Given that it originates from a very distantly-related species, the *S. pombe* DNA may not contain any such signals that would be recognisable to the mammalian cellular machinery. As such, it could be considered as a large stretch of 'inert' DNA over which heterochromatin would easily spread. It then remains to be determined from what source the heterochromatin could spread to cover the inserted DNA.

The surrounding mouse DNA is one potential source for heterochromatin spreading into the *S. pombe* DNA. However, this does not seem to be the case for the F1.1 cell line at least, in which the *S. pombe* DNA is inserted in an intergenic region in close proximity to two actively transcribed genes. Another possibility is that the cell reacts to the introduction of foreign DNA and proceeds to silence it in a manner analogous to its silencing of parasitic genetic elements by H3K9me3 (*Mikkelsen et al., 2007*; *Timms et al., 2016*; *Cuellar et al., 2017*). In this scenario, heterochromatin

would spread from one or several small nucleating regions of heterochromatin within the insert. However, this cannot occur in all cases as some cell lines transfected with *S. pombe* DNA do not form heterochromatin (*Figure 5G,H*). Perhaps this merely reflects the stochastic nature of foreign DNA silencing, so that a small number of inserts might escape the silenced fate of the majority of others. This raises the possibility that cell lines such as NP-T1 and NP-T2 in which we did not observe high H3K9me3 over the *S. pombe* DNA might acquire it upon continued growth in culture, and that 'intermediate' cell lines such as NP-T3 which show moderate H3K9me3 levels and occasional distinct structures may represent a transitional stage, or at least reflect this stochasticity in its cell-to-cell variability.

## Heterochromatin alters mitotic chromatin structure

In all mammalian-*S. pombe* cell lines examined the presence of heterochromatin correlates with a distinct appearance of the *S. pombe* DNA region on metaphase chromosomes (*Figure 5*). Engineering the removal of H3K9me3 from those insertions on which it is present would be a more direct method of determining its influence on the chromosome structure in these cell lines. However, the challenge here lies in finding a method of doing so both selectively and without disrupting cellular function excessively. Several drugs exist which act to remove heterochromatic signals from chromatin, however their targets are frequently varied and they affect all heterochromatin, and sometimes even non-heterochromatic post-translational modifications, rendering conclusions as to their effect more difficult to ascribe to a single factor (*Zheng et al., 2008*; *Tóth et al., 2004*; *Cherblanc et al., 2013*). In addition such drugs, as well as most mutants which significantly affect heterochromatin, are also either lethal or hinder progress through the cell cycle, making analysis of mitotic features difficult. The establishment of a technique by which the H3K9me3 could be more selectively or unobtrusively removed from the *Sp-DNA* insert would therefore be a powerful tool to develop for future investigations.

Nevertheless, the strength of the correlation across the multiple and varied cell lines investigated points to a likely link between heterochromatin and the distinct mitotic structure observed. No other factor examined in this study fully correlates with the changes in appearance of the chromatin. Neither the host cell line nor the method of DNA insertion (fusion or transfection) correlated with the distinct structure observed. The source or sequence of the DNA is unlikely to be a determining factor given that all *S. pombe* DNA inserts were derived from the same *S. pombe* strain (with the exception of the *clr4Δ* strain which differed only in a single gene deletion). Nor did the size or relative chromosomal position of the insert seem to matter. Both NP-T1 and NP-T4 cells have *S. pombe* DNA insertions of a similar size, but only the *S. pombe* DNA insert in NP-T4 exhibits a distinct heterochromatin-associated distinct structure (*Figure 5A,J*). Conversely, both NP-T2 and NP-*clr4Δ*-F4 cells have comparatively large insertions close to centromere regions but only NP-*clr4Δ*-F4 cells exhibit a distinct structure on the inserted DNA (*Figure 5B* and *Figure 4A*). The presence of H3K9me3 not only correlated with the presence of a distinct mitotic chromatin structure, the magnitude of the two observations also showed correlation, as the inserts with the highest levels of H3K9me3 also displayed the most evident structure (*Figure 5*). This places H3K9me3-heterochromatin as the primary candidate for causing these distinct chromatin structures.

This raises the question of whether heterochromatin is involved in the regulation of chromosome structure at endogenous loci. Certainly, the most obvious example of a distinct chromatin structure, the centromere, is associated with large domains of pericentromeric H3K9me3 heterochromatin (*Kim and Kim, 2012*). Reports of condensin enrichment at centromeric regions also support a link between the structure of centromere regions and that seen over *S. pombe* DNA inserts, as this would match with our findings of the increased condensin:chromatin ratio (*Wang et al., 2005*; *D'Ambrosio et al., 2008*, *Kim et al., 2013*, *Figure 6H,I*, *Figure 6—figure supplement 2*). However, in the cell lines described here the centromere and *S. pombe* DNA insert appear to adopt different structures both from each other and the rest of the chromosome in mitosis. Centromeres appear as a region of increased DNA staining instead of the decrease observed over the *S. pombe* DNA insert, possibly indicating greater chromatin compaction (*Figure 1*). On the other hand, centromeres do appear as constrictions on mitotic chromosomes, which would be consistent with their organisation into smaller chromatin loops, consistent with our analyses. Furthermore, in the HeLa-*S. pombe* hybrid cell HeP-F3 the centromere and *S. pombe* insert did bear greater resemblance than in the

mouse hybrid cells, in which the centromeres are acrocentric, potentially complicating the comparison (*Figure 2C*, *Figure 2—figure supplement 1C*).

Further analyses in these and other species could perhaps determine with more certainty if certain structural properties are shared between the regions of H3K9me3 heterochromatin over *S. pombe* DNA and centromeric regions. It is possible that centromeres combine aspects of the smaller loop sizes we observe and increased compaction of the chromatin fibre. The effect of H3K9me3 on chromosome structure may be combined or in competition with others at centromeres, where a number of complex processes related to kinetochore function are at work (*Verdaasdonk and Bloom, 2012*). The influence of H3K9me3 heterochromatin on mitotic chromosome structure is thus potentially one of many factors that contributes to the distinct structure of centromeres in mitosis.

## Increased loading of condensin on heterochromatin and smaller chromatin loop arrays

While the role of H3K9me3-heterochromatin in compacting the 10 nm chromatin fibre in interphase is well known, its involvement in mitotic chromatin structure is less well explored. Histone modifications have been proposed to be a contributor to mitotic chromosome structure, potentially involved in the 'compaction' of chromatin by processes independent of the so-called 'shaping' action of organising proteins such as the condensin complexes (*Zhiteneva et al., 2017*). H3K9me3 in particular has been shown to increase during mitosis, and a role for it in chromosome compaction has been put forward (*McManus et al., 2006*, *Park et al., 2011*). However, the distinct structure presented here, characterised by a decreased rather than increased chromatin density, is at odds with the idea that heterochromatin is compacting the chromatin fibre and rather suggests a role for heterochromatin in the 'shaping' of chromatin by condensins. This is supported by our analysis of the structure, which points to an altered organisation of the *S. pombe* DNA into smaller loops of chromatin than the surrounding mouse chromatin (*Figure 6*). These results suggest a role for H3K9me3-heterochromatin in the regulation of condensin activity that leads to this altered organisation.

Shaping of mitotic chromosomes is thought to be achieved by the ATPase activity of the condensin complexes, which extrude loops of chromatin through their ring-like structures (*Goloborodko et al., 2016*; *Hassler et al., 2019*). This shaping is thus mediated by a small number of parameters, including the frequency of condensin loading onto and dissociation from the chromatin fibre, and the processivity of condensin-mediated extrusion. Given the different roles of condensin I and II in the shaping of mitotic chromosomes, the relative amount of these two complexes also has an impact on chromosome structure (*Shintomi and Hirano, 2011*; *Green et al., 2012*). The dynamic activity of the different condensin complexes thus reaches an equilibrium, and formation of loops-within-loops ensures the stability of the final structure (*Goloborodko et al., 2016*; *Gibcus et al., 2018*). Alteration in any of these parameters would result in an alteration of the equilibrium and thus a change in the final structure. Thus, a local alteration in the activity of the condensin complexes could explain the altered organisation around the *S. pombe* DNA that results in the observed distinct structure.

In order to account for all of our observations, we propose a model whereby condensin is enriched at heterochromatin by a localised increase in its initial loading (*Figure 7*). This increase in condensin loading results in an array of smaller loops than non-heterochromatic regions because the greater concentration of condensin complexes may block or slow each other's loop extrusion activity. Thus, the loops are not free to extrude to the extent they would be otherwise and an alternative equilibrium with smaller loops is attained. While maintaining a similar self-organising scaffold of condensin, the resulting structure has on average less chromatin being tethered by each condensin complex, meaning that the condensin:chromatin ratio increases, the amount of chromatin per unit length decreases and the diameter of the resulting region of the chromosome is narrower, explaining all of our observations (*Figure 7C*).

While we put forward this model as that which best explains our results, there are many additional aspects of this experimental system that warrant further study. The mechanism by which heterochromatin might lead to increased condensin recruitment is intriguing, and one which bears relevance to the study of the role of condensin at centromeres (*Wang et al., 2005*; *D'Ambrosio et al., 2008*, *Kim et al., 2013*). The specific removal of heterochromatin from *S. pombe* DNA inserts would be a direct way to test the role of heterochromatin in altering mitotic chromosome architecture in these cell lines, and merits further investigation. The contribution of sequence

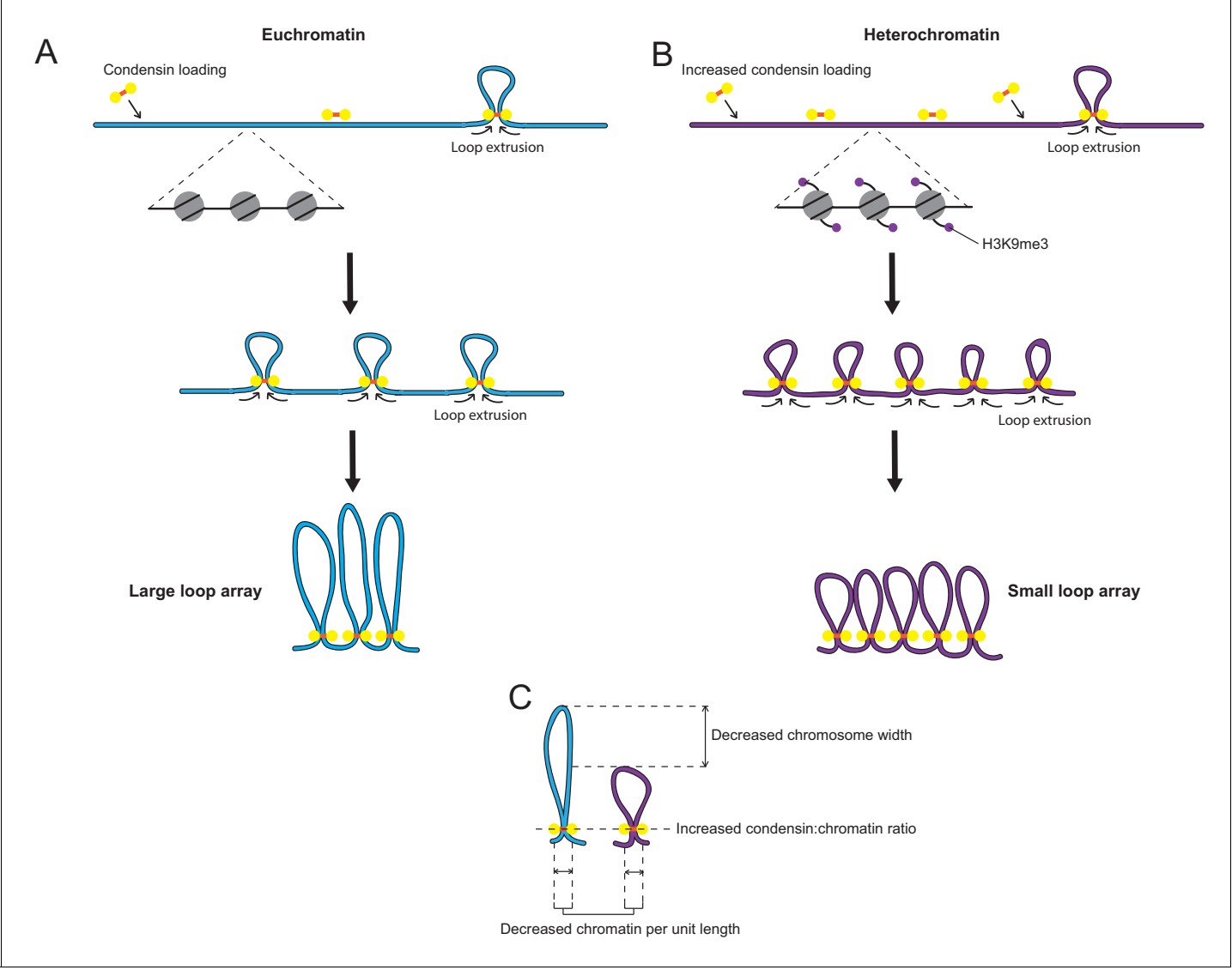

**Figure 7.** Model: Increased loading of condensin on heterochromatin leads to smaller chromatin loops on the mitotic chromosome. (A) Diagram of regular chromatin organisation by loop extrusion in mitosis. Condensin binds to the chromatin fibre and extrudes loops until blocked by other condensin complexes on either side. This results in an array of large side-by-side loops organised around a condensin scaffold. (*Goloborodko et al., 2016*; *Gibcus et al., 2018*). (B) When condensin loading is increased over heterochromatin loop extrusion is halted earlier by the greater concentration of condensin complexes present on the chromatin fibre. This results in an array of smaller loops. (C) This small-loop array results in a decrease in chromatin per unit length, an increase in the ratio of condensin to chromatin and a narrowing of the chromosome width.

features to the structure of the foreign DNA is another potential avenue of future research. While we place more importance on the epigenetic factors due to the source of the inserted DNA being the same in all our cell lines, it would be of interest to determine whether distinct structures also result from the insertion of DNA from other sources with widely differing GC content. The lower frequency of predicted binding sites suggests that CTCF is unlikely to be involved in shaping mitotic chromosome architecture across *Sp-DNA* inserts, however, the experimental manipulation of CTCF-binding site frequency would definitively test if CTCF and cohesin influence loop formation during mitosis (*Costantino et al., 2020*). In general, the cell lines generated in this study provide an excellent system for studying large domains of heterochromatin in the absence of confounding factors such as repetitive sequences and may provide further insight into the role of heterochromatin at naturally occurring regions such as centromeres.

# Materials and methods

**Key resources table**

| Reagent type (species) or resource | Designation | Source or reference | Identifiers | Additional information |
|---|---|---|---|---|
| Cell line (*M. musculus*) | C127 (female, mammary tumour) | ATCC | CRL-1804; | |
| Cell line (*M. musculus*) | F1.1 | *Allshire et al., 1987* | | Kept as cryopreserved stocks in the Allshire lab |
| Cell line (*M. musculus*) | NIH3T3 | ATCC | CRL-1858 | |
| Cell line (*H. sapiens*) | HeLa | ATCC | CCL-2 | |
| Cell line (*M. musculus*) | NP-F1 | This paper | | NIH3T3-*S. pombe* fusion cell line. Aka NPA4. Available from Allshire lab. |
| Cell line (*M. musculus*) | NP-F2 | This paper | | NIH3T3-*S. pombe* fusion cell line. Aka NPF3-19. Available from Allshire lab. |
| Cell line (*M. musculus*) | NP-T1 | This paper | | NIH3T3-*S. pombe* DNA transfection cell line. Aka NPD3. Available from Allshire lab. |
| Cell line (*M. musculus*) | NP-T2 | This paper | | NIH3T3-*S. pombe* DNA transfection cell line. Aka NPD5. Available from Allshire lab. |
| Cell line (*M. musculus*) | NP-T3 | This paper | | NIH3T3-*S. pombe* DNA transfection cell line. Aka NPT-C606. Available from Allshire lab. |
| Cell line (*M. musculus*) | NP-T4 | This paper | | NIH3T3-*S. pombe* DNA transfection cell line. Aka CPT-C500. Available from Allshire lab. |
| Cell line (*M. musculus*) | NP-T5 | This paper | | NIH3T3-*S. pombe* DNA transfection cell line. Aka NPT-C480. Available from Allshire lab. |
| Cell line (*M. musculus*) | NP-T6 | This paper | | NIH3T3-*S. pombe* DNA transfection cell line. Aka NPT-C482. Available from Allshire lab. |
| Cell line (*M. musculus*) | NP-*clr4Δ*-F4 | This paper | | NIH3T3-*S. pombe* fusion cell line. Aka NPF-Clr-I. Available from Allshire lab. |
| Cell line (*H. sapiens*) | HeP-F3 | This paper | | HeLa-*S. pombe* fusion cell line. Aka HeP3. Available from Allshire lab. |
| Strain, strain background (*S. pombe*) | FY43 | *Allshire et al., 1987* | | *h-Int5(pUraSV2Neo) ade6-210 leu1-32 ura4-D18* |
| Antibody | Anti-H3K9me3 (rabbit polyclonal) | Active Motif | 39161; RRID:AB_2532132 | ChIP(4 µg/mL) IF(1:1000) |
| Antibody | Anti-H3K4me3 (rabbit polyclonal) | Active Motif | 39159; RRID:AB_2615077 | ChIP(4 µg/mL) |
| Antibody | Anti-H3K36me3 (rabbit polyclonal) | Active Motif | 61101; RRID:AB_2615073 | ChIP(4 µg/mL) |
| Antibody | Anti-H3K9ac (rabbit polyclonal) | Active Motif | 39137; RRID:AB_2561017 | ChIP(4 µg/mL) |

*Continued on next page*

*Continued*

| Reagent type (species) or resource | Designation | Source or reference | Identifiers | Additional information |
|---|---|---|---|---|
| Antibody | Anti-H3K27me3 (rabbit polyclonal) | Abcam | Ab4729; RRID:AB_2118291 | ChIP(4 µg/mL) |
| Antibody | Anti-HP1alpha (goat polyclonal) | Abcam | Ab77256; RRID:AB_1523784 | ChIP(4 µg/mL) IF(1:500) |
| Antibody | Anti-SMC2 (rabbit polyclonal) | Losada Lab | | IF(1:1000); Obtained from Ana Losada |
| Antibody | Anti-SMC3 (rabbit polyclonal) | Losada Lab | | IF(1:1000); Obtained from Ana Losada |
| Antibody | Anti-H4 (rabbit polyclonal) | Sigma-Aldrich | SAB4500312; RRID:AB_10743081 | IF(1:1000) |
| Antibody | Anti-CAPH1 (rabbit polyclonal) | Abcam | Ab154105 | IF(1:1000) |
| Antibody | Anti-rabbit-A594 (donkey polyclonal) | Invitrogen | A21207; RRID:AB_141637 | IF(1:10000) |
| Antibody | Avidin-FITC | Vector Labs | A-2001–5; RRID:AB_2336455 | FISH(1:500) |
| Antibody | Biotinylated anti-Avidin (goat polyclonal) | Vector Labs | BA-0300-.5; RRID:AB_2336108 | FISH(1:100) |
| Antibody | Texas Red anti-sheep (rabbit polyclonal) | Vector Labs | TI-6000; RRID:AB_2336219 | FISH(1:100) |
| Antibody | Rhodamin anti-dig (sheep polyclonal) | Roche | 11207750910; RRID:AB_514501 | FISH(1:20) |
| Recombinant DNA reagent | H2B-GFP | Addgene | 11680 | |
| Commercial assay or kit | EpiTect Bisuflite kit | Qiagen | 59104 | |
| Chemical compound, drug | LightCycler 480 SYBR Green Master Mix | Roche | 04707516001 | |
| Software, algorithm | FIMO | *Grant et al., 2011* | | |
| Software, algorithm | HiCExplorer | *Ramírez et al., 2018* | | |

## Cell culture, fusion and transfection

C127, NIH3T3 and HeLa cells were from ATCC. All mouse and human cell lines were maintained in DMEM supplemented with 10% FBS and 100 U/mL Penicillin-Streptomycin and 400 µg/mL geneticin (Gibco, Carslbad, CA). Standard *S. pombe* culture methods were followed as described (*Moreno et al., 1991*). Cells were grown in YES at 32℃. For protoplasting of *S. pombe* a procedure modified from previously reported methods was used (*Allshire et al., 1987*; *Flor-Parra et al., 2009*). Briefly, *S. pombe* cells were harvested from log-phase cultures grown in YES. $1 \times 10^9$ cells were resuspended in 10 mL SP2 buffer (50 mM citrate-phosphate, 1.2 M Sorbitol, pH 5.6) containing 25 mg/mL Lallzyme (Lallemand, Rexdale, ON, Canada, supplied by Litmus Wines), then incubated at 36℃ for approximately 45 min until 80–90% of cells were converted to spherical protoplasts (determined by phase contrast microscopy). Protoplasts were pelleted at 1600xg for 3 min, then gently resuspended in 5 mL wash buffer (1.2 M Sorbitol, 10 mM Tris-HCl pH 7.5) and re-pelleted a total of three times.

For fusion of protoplasts with mammalian cells, protoplasts were counted using a haemocytometer and $1 \times 10^8$ cells were aliquoted per fusion. Mammalian cells were trypsinised, collected, washed in serum-free DMEM (SF-DMEM), centrifuging at 300xg for 5 min, and similarly counted. $1 \times 10^7$ mammalian cells in 5 mL SF-DMEM were then added to the tube containing the *S. pombe* protoplast pellet without dislodging it and pelleted on top at 300xg for 5 min. Fusion was carried out by resuspending the mixed cell pellet slowly in 1 mL of 50% PEG 1500 (Roche, Basel, Switzerland) at 37℃ over the course of 1 min and incubated for a further 1 min at 37℃. The total volume was

brought to 5 mL with SF-DMEM gradually over 4 min, and then to 15 mL adding SF-DMEM drop by drop. Cells were then incubated for a further 5 min at 37°C before centrifugation and washing twice in SF-DMEM. Cells were finally resuspended in the appropriate cell culture media and plated.

A plasmid expressing H2B-GFP was obtained from Addgene (plasmid #11680) and transfected using the Neon transfection system (Invitrogen, Carlsbad, CA) as per the manufacturer's instructions, electroporating for two pulses at 1400V for 20 ms each. For the transfection of *S. pombe* DNA into mammalian cells, DNA was prepared using a Qiagen Blood and Cell Culture DNA Kit (Qiagen, Venlo, Netherlands). 15–20 µg DNA was added to 124 µL of 2M $CaCl_2$ and the total volume brought to 1 mL. 1 mL 2xHBS (50 mM HEPES pH 7.05, 10 mM KCl, 12 mM dextrose, 280 mM NaCl, 1.5 mM $Na_2PO_4$) was then added drop-wise while aerating constantly. 1 mL of the final mix was added drop-wise to a 10 cm diameter plate with cells at 30–40% confluence, or relatively equivalent volumes to other sized plates.

## FISH

For FISH metaphase spreads were prepared from non-arrested cells collected by mitotic shake-off. Cells were centrifuged at 800xg for 8 min and resuspended drop by drop in 5 to 10 mL 100 mM KCl pre-warmed to 37°C, mixing continuously. Cells were then incubated at 37°C for 15 min, centrifuged again and resuspended in 5 to 10 mL fixing solution (3:1 mix of methanol: acetic acid) at −20°C. Cells were centrifuged and washed in fixing solution in the same way twice more and finally resuspended in an appropriate volume of fixing solution (depending on cell number). 10 µL of fixed cell suspension was dropped onto a glass slide from approximately 0.5 m and allowed to dry for 2 to 5 days before proceeding. Slides were then incubated in 2xSSC with 100 µg/mL RNase at 37°C for 1 hr then dehydrated through a series of 2 min washes in 70%, 90% and 100% ethanol and air dried. Slides were then denatured in 2xSSC, 70% formamide, pH 7.5 for 1.5–2 min at 70°C. Denaturation was rapidly stopped by dipping slides into 70% ethanol on ice, followed again by dehydration through 90% and 100% ethanol at room temperature and air drying. Biotin or digoxigenin labelled *S. pombe* gDNA probes made by nick translation were evaporated at 65°C in 70% ethanol and resuspended in hybridisation mix (2xSSC, 50% deionised formamide, 10% dextran sulfate, 1% Tween 20). 100–200 ng probe containing 5 µg sonicated salmon sperm DNA in 15 µL hybridisation mix was used per slide. Alternatively, mouse chromosome paint was used directly. Probes were denatured at 70°C for 5 min and snap-cooled on ice before being placed on the slide under a coverslip overnight at 37°C in a humidity chamber. The following day slides were washed four times for 3 min in 2xSSC at 45°C and four times for 3 min in 0.1xSSC at 60°C before a brief wash in 4xSSC, 0.1% Tween 20 at 37°C. Blocking was performed for 5 min in in 4xSSC, 5% milk powder. Slides were then incubated with the appropriate detection antibodies for 30 to 60 min in 4xSSC, 5% milk powder under a coverslip. After each antibody incubation slides were washed three times for 2 min in 4xSSC, 0.1% Tween 20 at 37°C. After all antibody incubations slides were mounted directly in mounting medium and sealed. Antibodies used for detection of biotinylated probes were Avidin-FITC (Vector Labs, Burlingame, CA, 1:500), biotinylated α-Avidin (Vector Labs, 1:100) and Avidin-FITC again; and for detection of DIG-labelled probes, rhodamine α-DIG Fab fragments (Roche 1:20) followed by Texas Red α-sheep antibody (Vector Labs, 1:100).

## Immunolocalisation

Spreads that were used for immunolocalisation experiments were prepared using a Shandon Cytospin (Thermo Scientific, Waltham, MA). Cells collected by mitotic shake-off, centrifuged, washed in D-PBS (Gibco) and resuspended at $3 \times 10^4$ cells/mL in 100 mM KCl pre-warmed to 37°C. Cell suspensions were then incubated for 15 min at 37°C and 100 µL of cells were spun onto glass slides using a Cytospin centrifuge at 1,800 rpm for 10 min. The slides were then incubated in a coplin jar for 10 min at room temperature in KCM buffer (120 mM KCl, 20 mM NaCl, 10 mM Tris-HCl pH 8.0, 0.5 mM EDTA, 0.1% Triton X-100) before blocking in 1% BSA in KCM buffer for 30 min at 37°C. Slides were then incubated in primary and secondary antibodies in blocking solution for 30 min and 45 min respectively at 37°C. After each antibody incubation the slides were washed twice in KCM buffer for 5 min. The slides were then fixed in 4% PFA in KCM for 10 min at 37°C and washed briefly once in KCM and twice in water before mounting and sealing. Antibodies used were as follows: rabbit α-H3K9me3 (Active Motif, Carlsbad, CA #39161), rabbit α-SMC2 and α-SMC3 (from Ana Losada),

rabbit α-H4 (Sigma-Aldrich, St Louis, MO, #SAB4500312), rabbit α-CAPH1 (Abcam, Cambridge, UK, #Ab154105), donkey α-rabbit Alexa594 (Invitrogen, #A21207). Slides were imaged and positions on the slide recorded before proceeding to FISH. FISH was carried out as indicated above, except that slides were denatured at 80°C for 20 min.

## Microscopy

Most microscope images were acquired on a DeltaVision Core system (Applied Precision, Issaquah, WA) with an Olympus UPlanSApo × 100 oil immersion objective and an LED light source. Camera (Photometrics Cool Snap HQ), shutter and stage were controlled through Softworx (Applied Precision). Other images were acquired on an Eclipse Ti2 (Nikon, Tokyo, Japan) using the NIS-Elements software for instrument control and image acquisition. Z-series were collected and subsequently projected into a single image using either Softworx or ImageJ software (National Institutes of Health, Bethesda, MD). For intensity measurements DeltaVision images were also deconvolved in Softworx. Intensity and chromosome length measurements were taken using ImageJ. Intensity measurements were then binned as a function of their relative position along the chromosome and averaged. Regions of the average chromosome profile with equal numbers of measurements were compared using the Kolmogorov-Smirnoff test. The regions to test corresponding to the site of *S. pombe* DNA were defined as collections of two to five contiguous bins centred on the bin with the highest FISH intensity value, and containing only those bins whose FISH intensity was equal to or greater than 70% of this maximum value. Control regions were then defined as a separate collection of contiguous bins of the same size as the test region and whose IF or DNA intensity values fell within one standard deviation of the overall mean of the dataset, thus excluding extreme regions such as the centromere. Wherever possible, two distant and non-overlapping control regions were selected.

## ChIP-qPCR

For ChIP, mammalian cells were grown to near confluence in tissue culture dishes before fixation in growth medium with 1% PFA for 10 min at room temperature. Fixation was stopped by addition of glycine to a final concentration of 125 mM for 5 min at room temperature before washing twice in ice cold PBS. Cells were then harvested in PBS by scraping and spun down at 800xg for 5 min at 4°C. The fixed cells were then washed once each in 5 mL Wash Buffer 1 (0.25% Triton X-100, 10 mM EDTA, 0.5 mM EGTA, 10 mM HEPES) and Wash Buffer 2 (200 mM NaCl, 10 mM EDTA, 0.5 mM EGTA, 10 mM HEPES), with a 10 min incubation step on ice after each resuspension.

Cells were then resuspended in between 0.3 and 0.6 mL of Lysis Buffer (1% SDS, 10 mM EDTA, 50 mM Tris pH 8.1, 0.1 mM PMSF, 1x Protease Inhibitor) and sonicated on a Diagenode Bioruptor for 15 min on a 30 s on/off cycle to obtain fragments between 400 and 600 bp. The sonicated chromatin samples were then centrifuged at 13,000 rpm for 10 min and 4°C, collecting the supernatant to clarify the lysate. 100 μL aliquots in lysis buffer were made containing 25 to 50 μg of chromatin for histone modification IPs and 100 μg for HP1α, as measured by Nanodrop (Thermo Scientific), before adding 900 μL of ChIP Dilution Buffer (167 mM NaCl, 16.7 mM Tris pH 8.1, 1.2 mM EDTA, 1.1% Triton X-100, 0.01% SDS). 10 μg inputs in 50 μL Lysis Buffer were also taken and stored at 4°C. 4 μg of the appropriate antibody was then added to the samples, including a no-antibody control for each ChIP, which were then shaken gently overnight at 4°C. Antibodies used were: α-H3K9me3 (Active Motif #39161), α-H3K4me3 (Active Motif #39159), α-H3K36me3 (Active Motif #61101), α-H3K9ac (Active Motif 39137), α-H3K27ac (Abcam #ab4729), α-HP1α (Abcam #Ab77256).

50 μL of pre-washed protein G Dynabeads were aliquoted for each sample, and the supernatant removed with the aid of a magnetic rack. The samples were then added to the beads and shaken again at 4°C for 4 to 5 hr. The samples were then washed sequentially for 10 min at 4°C once each in RIPA (150 mM NaCl, 50 mM Tris pH 8, 0.1% SDS, 0.5% Na deoxycholate, 1% NP40), High Salt Wash (0.5 M NaCl, 50 mM Tris pH 8, 0.1% SDS, 1% NP40) and LiCl Wash (250 mM LiCl, 50 mM Tris pH 8, 0.5% Na deoxycholate, 1% NP40) and twice in TE Buffer (10 mM Tris pH 8, 1 mM EDTA). DNA was then eluted from the Dynabeads by addition of 400 μL Elution Buffer (2% SDS, 0.1 M NaCO3, 10 mM DTT) and shaking strongly for 30 min at room temperature. For this step and all subsequent steps the input was included with the IP samples. The eluate was then collected on the magnetic rack and the beads discarded. The DNA-protein crosslinks were then reversed by addition of 20 μL

4 M NaCl and shaking overnight at 300 rpm and 65°C. Proteins were then digested in 0.05 mg/mL proteinase K for 1 hr at 55°C, with addition of 8 μL 0.5 M EDTA, 16 μL 1 M Tris pH6.5.

DNA was then Phenol-Chloroform extracted and ethanol precipitated before RNase treatment in 0.25 μg/mL DNase-free RNase for 30 min at room temperature. DNA amounts were then determined by qPCR on a Light Cycler 480 using SYBRGreen qPCR master mix (Roche). Inputs were diluted 1:20 before qPCR. Enrichments were calculated as % of DNA immunoprecipitated relative to input at the locus in question, and normalised to a positive control locus. Normalised IPs were compared between loci using the Student's t-test.

## Bisulfite sequencing

Bisulfite conversion of mammalian genomic DNA was performed using the EpiTect Bisulfite kit (Qiagen). Selected targets of 200–300 bp were then amplified by Hot Start PCR using primers designed for converted DNA. PCR products were then gel extracted and cloned into a vector using the Strataclone PCR Cloning kit (Agilent, Santa Clara, CA) and Sanger sequenced using the BigDye Terminator Cycle sequencing kit (Applied Biosystems, Foster City, CA).

## DNA sequencing, assembly and analysis

High molecular weight F1.1 genomic DNA was sequenced on a PromethION flow cell (Oxford Nanopore) to generate ultra-long reads. Mean read length was approximately 7 kb with a N50 of 40 kb and the longest read reaching 300 kb in length. From this, the sequence of the *S. pombe* insert of F1.1 was assembled by minimap2 and miniasm (*Li, 2016*). The assembly was then polished using Illumina whole genome paired-end reads (*Langmead and Salzberg, 2012*; *Li, 2011*; *Li et al., 2009*). CTCF site prediction was performed using FIMO (*Grant et al., 2011*) with a value <0.0001. The CTCF motifs were based on position weight matrices downloaded from CTCFBSDB 2.0 (*Bao et al., 2008*; *Ziebarth et al., 2013*).

## Hi-C

F1.1 cells were synchronised for mitotic Hi-C by a thymidine block. Cells were arrested for 18 hr by the addition of 2 mM thymidine and collected by mitotic shake-off 5 and then 8 hr after release. Cells were fixed in 1% PFA and collected as for ChIP. Chromatin was collected and Hi-C libraries prepared as previously described (*Naumova et al., 2013*). Hi-C data were mapped to the mouse reference genome (GRCm38) and the *S. pombe* insert assembly using HiCExplorer (*Ramírez et al., 2018*) and distance decay plots were made using Cooltools.

## Acknowledgements

We thank W C Earnshaw, S Boyle, W Bickmore, H Mjoseng and R Meehan for advice and technical help as well as the lending of equipment. We also thank R Clark and L Murphy of the ECRF Genetics Core and the University of Nottingham Deep Seq facility for help with MinIon sequencing and A Losada for providing the anti-SMC2 antibody. Finally we thank D Kelly of the Wellcome Centre for Cell Biology Optical Imaging Laboratory for help with microscopy and image analysis and A Kerr of the Bioinformatics Core Facility for advice on statistical analysis. MHF-J was supported by the Wellcome 4 year PhD programme in Cell Biology (102336), JD by the National Human Genome Research Institute (HG003143) and RCA by a Wellcome Trust Principal Research Fellowship (095021 and 200885). Research was also supported by core funding for the Wellcome Centre for Cell Biology (203149). JD is an investigator of the Howard Hughes Medical Institute.

## Additional information

### Competing interests

Job Dekker: Reviewing editor, *eLife*. The other authors declare that no competing interests exist.

## Funding

| Funder | Grant reference number | Author |
| --- | --- | --- |
| Wellcome Trust | Wellcome 4 year PhD studentship | Maximilian H Fitz-James |
| Wellcome Trust | 102336 | Maximilian H Fitz-James |
| Wellcome Trust | Principal Research Fellowship | Maximilian H Fitz-James Pin Tong Alison L Pidoux Sharon A White |
| Wellcome Trust | 095021 | Maximilian H Fitz-James Pin Tong Alison L Pidoux Sharon A White Robin Allshire |
| Wellcome Trust | Principal Research Fellowship | Maximilian H Fitz-James Pin Tong Alison L Pidoux Sharon A White Robin C Allshire |
| Wellcome Trust | 200885 | Maximilian H Fitz-James Pin Tong Alison L Pidoux Sharon A White Robin C Allshire |
| Wellcome Trust | Wellcome Centre for Cell Biology Core grant | Maximilian H Fitz-James Pin Tong Alison L Pidoux Sharon A White Robin Allshire |
| Wellcome Trust | 203149 | Maximilian H Fitz-James Pin Tong Alison L Pidoux Sharon A White Robin Allshire |
| National Human Genome Research Institute | HG003143 | Hakan Ozadam Liyan Yang Job Dekker |

The funders had no role in study design, data collection and interpretation, or the decision to submit the work for publication.

## Author contributions

Maximilian H Fitz-James, Conceptualization, Data curation, Formal analysis, Investigation, Visualization, Writing - original draft, Writing - review and editing; Pin Tong, Data curation, Formal analysis, Visualization; Alison L Pidoux, Conceptualization, Investigation, Writing - review and editing; Hakan Ozadam, Data curation, Formal analysis; Liyan Yang, Sharon A White, Investigation; Job Dekker, Conceptualization, Resources, Supervision, Funding acquisition, Methodology, Project administration, Writing - review and editing; Robin C Allshire, Conceptualization, Resources, Supervision, Funding acquisition, Project administration, Writing - review and editing

## Author ORCIDs

Maximilian H Fitz-James http://orcid.org/0000-0002-6084-5887
Job Dekker https://orcid.org/0000-0001-5631-0698
Robin C Allshire https://orcid.org/0000-0002-8005-3625

## Decision letter and Author response

Decision letter https://doi.org/10.7554/eLife.57212.sa1
Author response https://doi.org/10.7554/eLife.57212.sa2

## Additional files

### Supplementary files
- Transparent reporting form

### Data availability

DNA sequencing and nanopore data were uploaded to the Sequence Read Archive with project ID PRJNA629899. Hi-C data was uploaded to GEO with accession ID GSE149677.

The following dataset was generated:

| Author(s) | Year | Dataset title | Dataset URL | Database and Identifier |
|---|---|---|---|---|
| Allshire R, Fitz-James MH, Ozadam H, Dekker J, Tong P | 2020 | Large domains of heterochromatin direct the formation of short mitotic chromosome loops | https://www.ncbi.nlm.nih.gov/geo/query/acc.cgi?acc=GSE149677 | NCBI Gene Expression Omnibus, GSE149677 |

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
