## [Decision Letter]

**Acceptance summary:**

This manuscript addresses an unusual and interesting observation initially made in 1987, showing that insertion of *S. pombe* yeast DNA into a mouse chromosome is associated with a visible 'thinning' of mitotic chromosome width. Here, the Allshire group use cutting-edge techniques to determine why the pombe chromatin assumes a centromere-like constriction that differs from the wider flanking mouse chromatin. They discover that enrichments for the heterochromatic histone modification H3K9me3, smaller loops and condensin complexes occur de novo upon introduction of either pombe DNA or chromatin into mammalian chromosomes. The authors propose that the characteristic constrictions associated with eukaryotic centromeres results from higher levels of condensin recruitment over heterochromatin, which produces smaller chromatin loops.

**Decision letter after peer review:**

Thank you for submitting your article "Large domains of heterochromatin direct the formation of short mitotic chromosome loops" for consideration by *eLife*. Your article has been reviewed by two peer reviewers, including Gary H Karpen as the Reviewing Editor and Reviewer #1, and the evaluation has been overseen by Jessica Tyler as the Senior Editor. The following individual involved in review of your submission has agreed to reveal their identity: Eric F Joyce (Reviewer #2).

The reviewers have discussed the reviews with one another and the Reviewing Editor has drafted this decision to help you prepare a revised submission.

Summary:

This manuscript addresses an interesting observation initially made in 1987 showing that insertion of *S. pombe* DNA into a mouse chromosome was associated with a visible 'thinning' of the mitotic chromosome width. Most biologists would posit that inserting even a large (Mb) size piece of DNA from one organism into a distantly-related organism would not impact chromosome structure. The authors generate a large number of independent cell lines that reproduce the constriction phenomenon, then use molecular genetic tools (e.g. ChIP, Hi-C) to conclude that H3K9me enrichments, smaller loops and condensin complex enrichment occur de novo upon introduction of pombe DNA or chromatin into mammalian chromosomes. One interesting result is the demonstration that the pombe DNA constriction does not depend on pre-existing chromatin state, since the phenotype is observed upon insertion of naked DNA.

Overall, reviewers agree that the manuscript contains interesting and important information that would be of significant interest to the field, warranting publication in *eLife* once the following issues are addressed.

Essential revisions for this paper:

The primary results are generally sound and rigorous. However, the manuscript would benefit from text revisions (not new experiments) that clarify approaches and rationales used to support some conclusions, and distinguish solid conclusions from speculation.

1) Make the rationale and evidence supporting differences in sizes of loops, vs. other forms of compaction, clear. An increase in interactions identified by Hi-C does not a priori indicate loop formation. Figure 6C seems to show a decrease in contact probability for *S. pombe* DNA at the smallest genomic distance near the y-axis. Is this consistent with the authors' conclusions? They highlight an extended increase followed by a large dip at the largest genomic distance to suggest that loop sizes are reduced and that this must be driven by condensin activity. However, loops defined by Hi-C are typically represented by corner peaks in contact matrices. Are corner peaks observed in this dataset? What is the evidence that these interactions are even loops?

2) It would be useful to know if there is a higher density of CTCF sites in the pombe DNA. Of course CTCF ChIP seq or Cut and Tag would be even better, but a motif search would be sufficient for now.

It would also be useful to know if there are other sequence features (e.g. AT richness) that distinguish naked pombe DNA from mammalian DNA. One other reason this should be examined is that DAPI is used here for DNA quantitation, and has known preferences for AT-rich DNA.

3) Using primary and secondary antibody labeling to measure fluorescence intensity is typically difficult to accurately quantify and compare across experiments. The authors normalize their numbers across samples by comparing intensities to neighboring chromatin regions, which is reasonable. However, it was not clear how these "control" regions were selected, and the size and location of control regions changed across different experiments and figures. It seems that in some cases the control regions are equally spaced from the insert across different conditions (as in Figure 1B, E, and F) but not in other cases (Figure 2D and E). Please clarify how control regions were selected.

4) With regard to the conclusion that condensin concentration is increased on the inserted DNA, it was not clear whether condensin I/II ratios would be expected to be changed. The authors state that "neither the overall levels of condensin nor the ratio of condensin I to condensin II complexes differ." It may be premature to make this conclusion without testing a condensin II-specific subunit as well. As the authors point out in the Introduction, different ratios can account for different chromosome structures. For instance, decreased lateral distances and increased axial distances, as observed at the inserted locus, could be explained by an increase in Condensin I:II ratio.

5) The model presented at the end is too complicated and should be simplified (and smaller) (PS condensin is mis-spelled (condensin) in the middle panel). Most importantly, the model is certainly valid, but the authors should consider and incorporate alternative models that could also account for the observed structural changes. The conclusions about K9me and condensin rely on seeing protein enrichments, but in the absence of direct perturbations a role for e.g. condensin is speculative. It may be worthwhile to consider and cite Costantino et al., 2020, where micro-C analysis of *S. cerevisiae* identified small loops in mitosis, in this case driven by cohesin. Other possibilities include differences in distributions of sequence features or binding sites (e.g. base composition or CTCF sites), and other possibilities mentioned above.

Revisions expected in follow-up work:

As with all interesting studies, these findings raise more questions and future experiments. Why is more condensin loaded onto these regions? What is the role of the RNAi or piRNA pathways in generating H3K9me on this 'foreign DNA'? Finally, perturbations are required to demonstrate that the constriction disappears upon H3K9 demethylation or blocking methylation, or depletion of condensin.

---

## [Author Response]

Essential revisions for this paper:The primary results are generally sound and rigorous. However, the manuscript would benefit from text revisions (not new experiments) that clarify approaches and rationales used to support some conclusions, and distinguish solid conclusions from speculation.1) Make the rationale and evidence supporting differences in sizes of loops, vs. other forms of compaction, clear. An increase in interactions identified by Hi-C does not a priori indicate loop formation. Figure 6C seems to show a decrease in contact probability for *S. pombe* DNA at the smallest genomic distance near the y-axis. Is this consistent with the authors' conclusions? They highlight an extended increase followed by a large dip at the largest genomic distance to suggest that loop sizes are reduced and that this must be driven by condensin activity. However, loops defined by Hi-C are typically represented by corner peaks in contact matrices. Are corner peaks observed in this dataset? What is the evidence that these interactions are even loops?

The reviewers are correct in saying that the Hi-C data does not a priori show loops but rather the drop in interaction frequency of the *P(s)* curve shows a decrease in DNA per layer of the mitotic chromosome consistent with an organisation into smaller loops. This data coupled with the observation that condensin is enriched is what led us to propose our model of decreased loop size.

To improve our discussion of this we have reworked our final Results section “Heterochromatinised *S. pombe* DNA forms smaller chromatin loops in mitosis” in order to better convey how this model was arrived at.

In response to the point concerning corner peaks in contact matrices: corner peaks are typically thought to be an indication of isolated loops and are generally only observed in interphase contact maps, indicating the boundaries of TADs. As mitotic Hi-C results represent an averaged profile of loop arrays which do not have fixed boundaries or anchor points, such features should not be visible in mitotic matrices and we would thus not expect to find them in our data.

2) It would be useful to know if there is a higher density of CTCF sites in the pombe DNA. Of course CTCF ChIP seq or Cut and Tag would be even better, but a motif search would be sufficient for now.It would also be useful to know if there are other sequence features (e.g. AT richness) that distinguish naked pombe DNA from mammalian DNA. One other reason this should be examined is that DAPI is used here for DNA quantitation, and has known preferences for AT-rich DNA.

We thank the reviewers for suggesting that we examine the predicted number of CTCF sites in the *S. pombe* genome vs. mouse DNA. These analyses have been added to Figure 6—figure supplement 2, with a brief discussion in the third paragraph of the subsection “Heterochromatinised *S. pombe* DNA forms smaller chromatin loops in mitosis”. We also included in this figure an examination the G+C content of the *S. pombe DNA* insert because it is also relevant to this comment.

These new analyses show that:

1) Predicted CTCF sites are actually reduced in frequency across the *S. pombe* DNA insert relative to the flanking mouse DNA. This suggests that less CTCF would associate with the *S. pombe* DNA inserts. If CTCF limits mitotic loop extrusion then larger rather than smaller loop arrays would be predicted to result from CTCF site reduced frequency across this DNA;

2) The G+C content of the *S. pombe* genome is 36% as opposed to 42% in the mouse genome. The reduced G+C content of the *S. pombe* genome must contribute to the reduced frequency of predicted CTCF binding, which is based on position weight matrices that heavily favour G and C.

Concerning the reviewers’ point about DAPI: It should be noted that in all experiments where DNA was quantified by microscopy propidium iodide not DAPI was used to stain DNA. This was done precisely to avoid the known AT-bias of DAPI and obtain an unbiased measurement of DNA density along chromosome lengths. A brief explanation of this rationale has also been added to the second paragraph of the subsection “*S. pombe* DNA incorporated into a mouse chromosome adopts a distinct structure with less DNA per unit length” to avoid confusion.

3) Using primary and secondary antibody labeling to measure fluorescence intensity is typically difficult to accurately quantify and compare across experiments. The authors normalize their numbers across samples by comparing intensities to neighboring chromatin regions, which is reasonable. However, it was not clear how these "control" regions were selected, and the size and location of control regions changed across different experiments and figures. It seems that in some cases the control regions are equally spaced from the insert across different conditions (as in Figure 1B, E, and F) but not in other cases (Figure 2D and E). Please clarify how control regions were selected.

The rationale behind the selection of test and control regions for statistical analysis of the chromosome profiles has been added to the subsection “Microscopy”. Briefly, the test *S. pombe* DNA regions were determined based solely on the FISH fluorescence intensity at the given points. Control regions were then selected to be regions on the same chromosome of equal size to the test region with IF or π values approaching the average for the whole chromosome. This choice was made to select control regions with an “average” signal and avoid artificially inflating the statistical significance. In cases where the distance between the regions being tested varies, as was pointed out, it is generally to avoid outlier regions that would in fact be even more statistically significant, but not necessarily representative (for instance the “bump” in Figure 2E in position 20-30).

However, upon re-inspection we have discovered that two of our graphs (Figure 5G and I) did not quite conform to this standard. The test and control regions have therefore been altered accordingly with no change to our conclusions, and we thank the reviewers for drawing our attention to this oversight.

4) With regard to the conclusion that condensin concentration is increased on the inserted DNA, it was not clear whether condensin I/II ratios would be expected to be changed. The authors state that "neither the overall levels of condensin nor the ratio of condensin I to condensin II complexes differ." It may be premature to make this conclusion without testing a condensin II-specific subunit as well. As the authors point out in the Introduction, different ratios can account for different chromosome structures. For instance, decreased lateral distances and increased axial distances, as observed at the inserted locus, could be explained by an increase in Condensin I:II ratio.

As the reviewers mention, measurement of a condensin II-specific subunit in addition to the condensin I-specific CAPH1 would be the ideal way of determining the ratio between the two complexes. Unfortunately, several attempts to do so with different antibodies against different condensin II subunits failed to yield any results on mitotic chromosome spreads in our hands. In the absence of this data however, we posit that it can be inferred that this ratio does not change if neither the overall levels of condensin, nor the levels of condensin I alone change. We have expanded our explanation of this with respect to our conclusion in the fifth paragraph of the subsection “Heterochromatinised *S. pombe* DNA forms smaller chromatin loops in mitosis”.

5) The model presented at the end is too complicated and should be simplified (and smaller) (PS condensin is mis-spelled (condensin) in the middle panel). Most importantly, the model is certainly valid, but the authors should consider and incorporate alternative models that could also account for the observed structural changes. The conclusions about K9me and condensin rely on seeing protein enrichments, but in the absence of direct perturbations a role for e.g. condensin is speculative. It may be worthwhile to consider and cite Costantino et al., 2020, where micro-C analysis of *S. cerevisiae* identified small loops in mitosis, in this case driven by cohesin. Other possibilities include differences in distributions of sequence features or binding sites (e.g. base composition or CTCF sites), and other possibilities mentioned above.

We have substituted the original model presented with a new simplified version. This new model now serves to compare the differences between the “natural” structure of chromatin and that expected due to the increased recruitment of condensin on inserted *S. pombe* DNA, as well as to illustrate how this model can account for our experimental observations. The discussion of this model (subsection “Increased loading of condensin on heterochromatin and smaller chromatin loop arrays”) has been changed accordingly and we now make reference to alternative possibilities and unexplored contributing factors. We agree that this simpler model makes our overall conclusions clearer and that our revised Discussion now suggests other possibilities and avenues for future research. We considered the possibility that CTCF site frequency might alter mitotic chromosome architecture but our finding that predicted CTCF site frequency is lower across the *S. pombe* DNA insert does not support this potential alternative model.